# Development of a micro-combined heat and power powered by an opposed-piston engine in building applications

Zhiming Gao [1] ✉, Philip Zoldak[2], Jacques Beaudry-Losique[2], Tony Mannarino[2], Jonathan Mansinger[2], Maysam Molana[2], Mingkan Zhang[1], Praveen Cheekatamarla[1], Ahmed Abuheiba[1], Hailin Li[3], Brian Fricke[1] & Kashif Nawaz[1]

Residential homes and light commercial buildings usually require substantial heat and electricity simultaneously. A combined heat and power system enables more efficient and environmentally friendly energy usage than that achieved when heat and electricity are produced in separate processes. However, due to financial and space constraints, residential and light commercial buildings often limit the use of traditional large-scale industrial equipment. Here we develop a micro–combined heat and power system powered by an opposed-piston engine to simultaneously generate electricity and provide heat to residential homes or light commercial buildings. The developed prototype attains the maximum AC electrical efficiency of 35.2%. The electrical efficiency breaks the typical upper boundary of 30% for micro–combined heat and power systems using small internal combustion engines (i.e., <10 kW). Moreover, the developed prototype enables maximum combined electrical and thermal efficiencies greater than 93%. The prototype is optimally designed for natural gas but can also run renewable biogas and hydrogen, supporting the transition from current conventional fossil fuels to zero carbon emissions in the future. The analysis of the unit's decarbonization and cost-saving potential indicate that, except for specific locations, the developed prototype might excel in achieving decarbonization and cost savings primarily in US northern and middle climate zones.

Residential homes and light commercial buildings usually require substantial heat and electricity simultaneously. Unfortunately, two-thirds of the energy used by conventional electricity generation is wasted in the form of heat discharged to the atmosphere[1], and additional energy is wasted during the transmission and distribution of electricity to end users. A combined heat and power (CHP) system is a high-efficiency energy technology that generates electrical power and captures heat that would otherwise be wasted, providing useful thermal energy–such as steam or hot water used for space heating and hot

water supply–in a single process and from a single energy source[2]. Thus, CHP systems enable more efficient and environmentally friendly energy usage than that achieved when heat and electricity are produced in separate processes. In fact, the exclusive features of CHP contribute to sustainable solutions for building decarbonization, especially the application of micro-CHP (mCHP) systems in single-family houses and multiple-family buildings that demand only a few kilowatts of electricity while requiring substantial space heating and water heating[3]. Thus, mCHP can be widely considered as a reliable and

[1]Oak Ridge National Laboratory, Oak Ridge, TN 37831, USA. [2]Enginuity Power Systems, 730 S Washington St, Alexandria, VA 22314, USA. [3]West Virginia University, Morgantown, WV 26506-6106, USA. ✉e-mail: gaoz@ornl.gov

decentralized system of heat and electricity production and can be installed as independent equipment at an immediate consumption location. Such distributed heat and electricity generation equipment using renewable energy is an excellent solution to reduce greenhouse gas emissions and to enhance the grid network security[4,5]. Substantial work has been performed to understand the economic benefit, marketing, and residential applications of mCHP in Asia[6–8], Europe[9,10], the Middle East[11], and North America[12–14]. There is a significant potential market demand for mCHP applications in the building sector.

Most mCHP systems are powered by internal combustion engines (ICEs), micro gas turbines, micro-Rankine cycles (ORC), Stirling engines, thermophotovoltaic generators (TPV), and fuel cells[3,13,15–28] (Supplemental Note 1). Fuel cell CHPs have the potential to achieve high electrical efficiency and low emissions, but significant challenges exist, such as the need for reliable and durable electrode materials, extra-slow start-up, and high cost[29–31]. Fuel cell CHP technology is still at a very early stage of development. Most commercial mCHPs are powered by ICEs and Stirling engines. Other technologies remain immature, have lower thermal efficiency, and are less cost-effective compared with ICE and Stirling engine technologies. Unlike ICE, the Stirling engine is an external combustion engine, in which heat is transmitted to the working fluid through an exchanger[32]. Commercial Stirling engines usually cannot achieve the efficiencies of ICEs because Stirling engines using low-cost materials cannot supply heat at 1,500–1,600 °C by thermal conduction. Therefore, ICE technologies remain the most attractive and well-established technology for the mCHP application[18,19,21]. Unfortunately, current ICE technologies in mCHP applications do not achieve more than 30% electrical efficiency, silent operation, low cost, or reduced maintenance[3]. Continuously developing novel technologies that improve the efficiency and reduce the cost of mCHPs is important for extending their market penetration.

Opposed-piston engines (OPEs) differ from conventional ICEs wherein the cylinder head is replaced by a second piston so that the two pistons in an OPE move toward each other during the compression stroke and away from each other during the expansion stroke. The architecture doubles the stroke-to-bore ratio. This technology enables a reduced displacement and a higher power density without excessive piston speed or exceeding the peak cylinder pressure limit. Compared with a conventional ICE, OPEs can reduce in-cylinder heat losses because the cylinder head is eliminated, and the combustion chamber has a lower surface-area-to-volume ratio[33]. These features enable a significant gain in thermal efficiency[34–36]. Moreover, OPEs are more cost-effective than conventional ICEs, mainly because they have 60% fewer parts per engine unit[37] and, therefore, have lower material and manufacturing costs. An OPE does not require a head gasket, large multiple head bolts, or a cylinder head. Consequently, OPEs are gaining interest in the automotive industry[33,36–38]. For example, Achates Power Inc[37,39]. has developed an opposed-piston two-stroke (OP2S) engine, which shows promise in the automotive industry. Achates claims that its OP2S engine designed for vehicles is 30%–50% more efficient than equivalent conventional petrol and diesel engines and 10% cheaper[37]. Enginuity Power Systems developed a gaseous-fuelled opposed-piston four-stroke (OP4S) engine and demonstrated up to 40% energy savings with boosted hydrogen operation[40,41]. Furthermore, OPEs have been widely applied as a propulsion system in the marine and aviation industries[42–44]. However, a literature search reveals that no mCHP systems have been powered by an OPE. The development of a compact OPE is critical as a major enabler of mCHPs because this technology can address the cost and efficiency challenges of existing technologies.

This work presents the development of a mCHP prototype powered by a small OP4S engine (i.e., <10 kW) to simultaneously provide heat and electricity to single-family houses or light commercial buildings. A light commercial building refers to a type of structure that is designed for smaller-scale-operation business and commercial purposes, such as retail stores, small offices, and restaurants[45]. The mCHP prototype designed for building applications aims to achieve cost-effective and flexible matching of thermal and electrical loads and aims to simplify distribution and installation processes while recovering and storing waste heat as hot water. The developed mCHP technology is expected to promote mCHP acceptance in the residential and light commercial markets because of its low cost and drop-in replacement feature using the building's existing connections. This is particularly important in remote and underserved communities that face significant challenges because of vulnerable energy infrastructure and a lack of access to reliable electricity. Such remote and underserved communities include rural villages in developing countries, mountainous regions, disaster-affected areas, and nomadic or transient communities.

## Results
### mCHP development
The mCHP unit consists of the Enginuity OP4S model V5.1, generator, rectifier, inverter, battery energy storage system, 52-gal water tank, and accessory components for hot water supply and space heating applications. The battery energy storage system is an external accessory component. The unit's module for engine control is an engine control unit (ECU), and its module for mCHP control is a programmable logic controller (PLC) to manage engine operation, battery charging, and waste heat recovery. Figure 1a, b show a verified prototype of the mCHP system with Technology Readiness Level (TRL) 6 that has been demonstrated in a laboratory operational environment. In the mCHP prototype, the OP4S can burn fuels to generate mechanical power and waste heat in a format of hot coolant and exhaust gas at the same time. The generated mechanical power drives the generator to produce AC electricity, which is used to meet the building's electricity demands or charge the battery after rectification to DC electricity through an onboard rectifier or enters the grid. The waste heat in the hot coolant and exhaust flow is recovered and stored in the 52-gal water tank, which is designed to connect to a building's hot water supply and space heating application. Figure 1c displays the mCHP system architecture and energy flow. The electricity and thermal energy outputs in this mCHP system can be adjusted by controlling the fuel flow rate and air-fuel ratio. Appropriate changes in spark timing of the OP4S engine will further affect the ratio of the electricity over thermal energy output available in exhaust gas while achieving reasonable engine efficiency. Moreover, the mCHP prototype is designed to connect the electrical battery module, which can flexibly deliver electricity by switching between various combustion modes to meet dynamic electrical and thermal energy demands of residential and light commercial buildings.

The mCHP can burn natural gas, propane, renewable biogas, and hydrogen. When considering global decarbonization goals, natural gas and propane are generally regarded as transitional or intermediate fuels rather than long-term sustainable options. However, global decarbonization will be a long-term effort. It is important for developing mCHP technologies to support the transition from current conventional fossil fuels to zero carbon emissions in the future. Thus, the proposed mCHP is optimally designed for natural gas but can also run renewable biogas and hydrogen. The anticipated timeline for the deployment of the proposed mCHP technology is targeted within 3–5 years.

The OP4S is a single-cylinder spark-ignition (SI) engine operating with premixed renewable natural gas or other gaseous fuels containing hydrogen. The configuration and specifications for the engine are shown in Fig. 2. In the OP4S, two pistons share one cylinder; each has its own crankshaft and connecting rod. The pistons move toward one another and meet at the top dead centre. As the pistons approach each other at the top of each stroke, the premixed natural gas is ignited by a

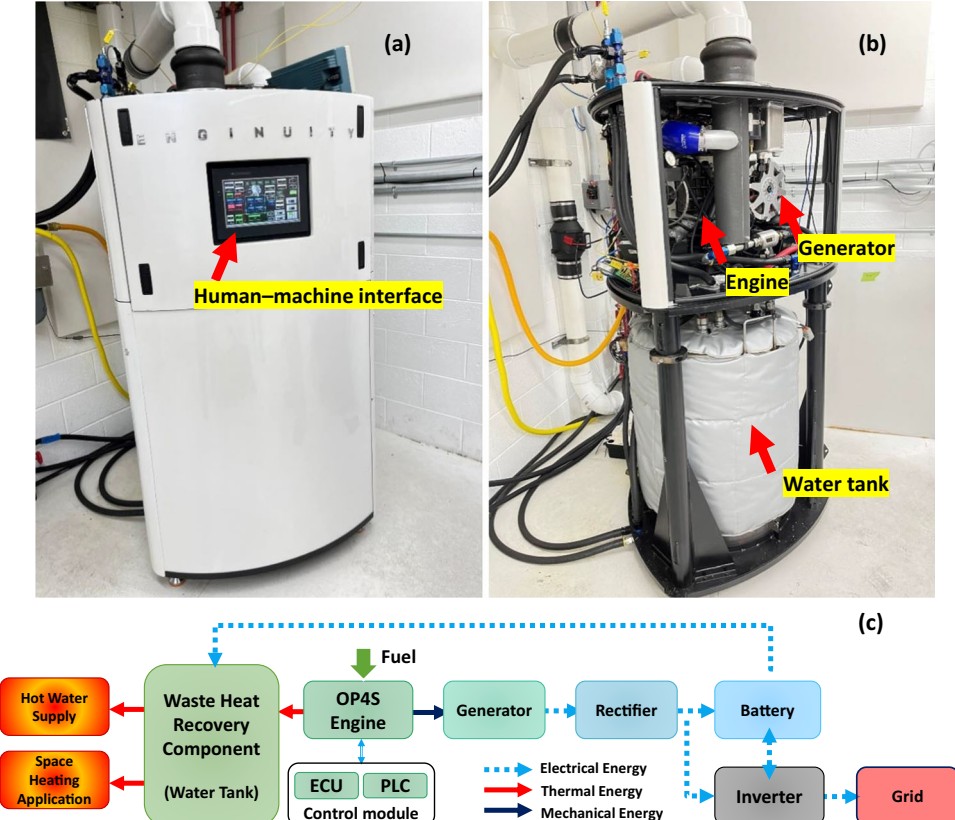

**Fig. 1 | The integrated mCHP system prototype in a testing lab. a** Shows the external view of the mCHP installed at our testing facility. **b** Exhibits the internal view of the mCHP. **c** Plots the mCHP system architecture and energy flow. Note: OP4S is opposed-piston four-stroke; ECU is engine control unit; and PLC is programmable logic controller.

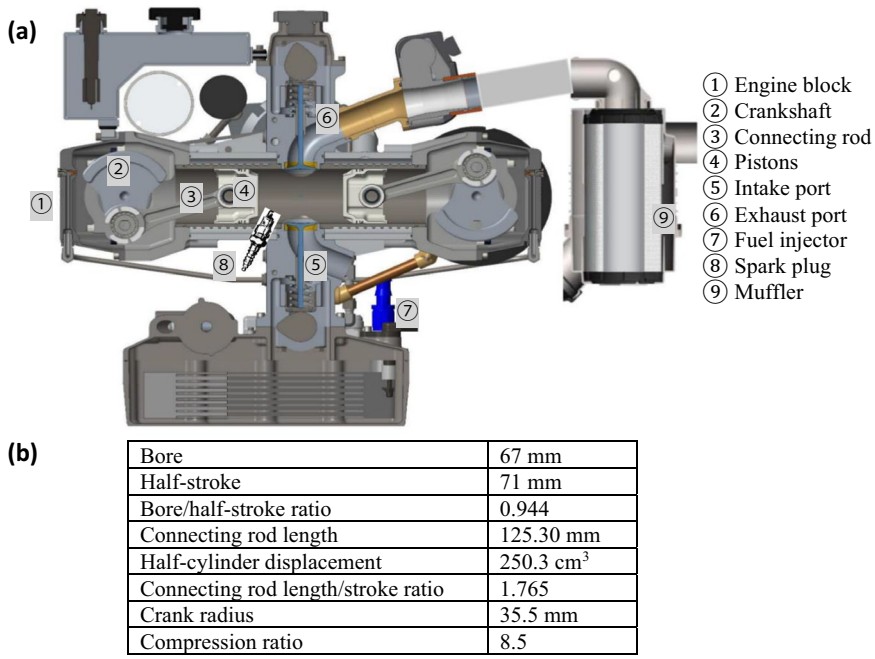

**Fig. 2 | The detailed configuration and specifications for the OP4S engine. a** Shows OP4S configuration marked with key components. **b** Lists key specifications of the OP4S. Note: OP4S is the abbreviation of opposed-piston four-stroke.

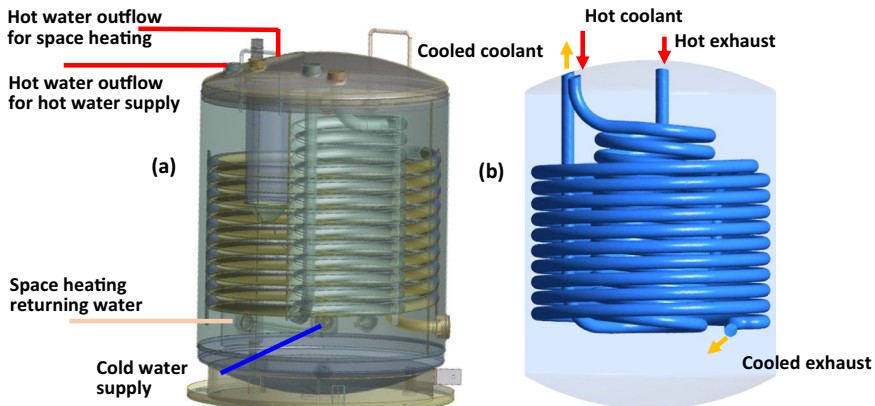

**Fig. 3 | mCHP waste heat recovery component. a** Shows the configuration of the mCHP waste heat recovery component labelled with hot water outflow, space heating return and cold water supply for space heating and hot water supply. **b** Displays the simplified architecture of the mCHP waste heat recovery component showing the helical coils for exhaust and coolant waste heat recovery, as well as inlets and outlets for hot coolant and exhaust.

spark plug in the cylinder, and combustion occurs, which converts fuel chemical energy to thermal energy, increases cylinder pressure, pushes the pistons apart, and produces mechanical work. The two crankshafts, one on each end of the engine, are joined by a set of gears from which mechanical power is transported to the generator and produces AC electricity. This engine operates on a four-stroke cycle with intake and exhaust flow controlled using intake and exhaust valves. The ECU and PLC control modules manage the engine operation under either the stoichiometric or lean combustion mode. Unlike conventional ICEs, which route the substantial heat of combustion to the cylinder head, the heat of combustion in the OP4S goes only to the opposing piston, reducing heat loss and increasing efficiency by 30%–50% more than that of comparable conventional petrol and diesel engines[37]. The OP4S engine is designed with a 20-year life span because of its excellent reliability and durability owing to substantially fewer parts than other engine types and low vibration[46].

In the mCHP, waste heat is recovered in two ways: by using the coolant flow circuit and by using the exhaust gas coil. To enable efficient engine operation and proper lubrication, the proposed mCHP includes a thermostat to maintain the returning coolant temperature around 70–80 °C to the engine by splitting the coolant flow from the engine into two separate streams: (1) part of the coolant flow enters the waste heat recovery component, and (2) the majority of the coolant flow bypasses the waste heat recovery component to mix with the cooled coolant from the waste heat recovery component. The recovered thermal energy is used to heat the water in the water tank and can be used as a domestic hot water supply and for space heating. Figure 3 shows the configuration of this mCHP waste heat recovery component, which includes a large-diameter helical coil for exhaust heat recovery and a small-diameter helical inner coil for waste heat recovery of the coolant. The water tank is designed to connect with two different electric heating elements—AC and DC electric heating—with two different power ratings. When the engine is off, the water tank can be heated by these electric heaters from the battery energy storage system or the external grid. The design allows the system to operate more efficiently while meeting flexible heat demands. This approach also significantly and cost-effectively lowers emissions.

Overall, deploying the mCHP technology in residential and light commercial buildings can provide uninterrupted heat and power at high efficiency and low cost without the constricts of severe weather. This is important for cold-climate regions and remote communities, which frequently experience severe weather or weather-related disasters. The mCHP can also be coordinated with intermittent renewable energy sources (e.g., wind and solar) to provide power and heat when the renewable energy sources are not available or during extended grid outages. It can also serve as the backbone for renewable energy–based microgrids by providing a reliable baseload source of electricity and thermal energy to support renewable energy resources and energy storage. By utilizing clean fuels such as hydrogen, the mCHP can achieve near-zero carbon emissions. In addition, the mCHP can be installed directly in buildings. Therefore, the mCHP is a compelling option for those seeking an uninterrupted, efficient, and cost-effective solution for both electricity and heat generation in residential and light commercial buildings.

## mCHP performance testing and evaluation

The developed mCHP prototype can provide up to 12 AC kW of electrical power. However, the control module limits its maximum power at 8.0 AC kW under the stoichiometric combustion mode by considering the safety, reliability, and durability of the prototype OP4S, and the control module limits its maximum power at 6.0 AC kW under the lean combustion mode. In this mode, the control module used the lambda sensor and intake port actuator to maintain a 30% excess of air (or $\lambda = 1.3$), and the ignition timing for all lean modes was advanced by ~10°CA to maximize engine brake thermal efficiency compared with the stoichiometric modes. Six and four testing cases were conducted to evaluate the mCHP performance under the stoichiometric and lean combustion modes, respectively (Supplemental Note 2). In the tests, the measurement uncertainty and testing repeatability were analysed. The results show that the uncertainties of electrical, thermal, and total CHP efficiency are 3.28%, 2.26%, and 2.51%, respectively (see Supplemental Note 3). The sensitivity comparison of the repeated tests for a 5.93 AC kW lean combustion mode was also conducted (see Supplemental Note 4), and the observation shows <1.8% error for all the performing efficiencies, power, OP4S exhaust, and coolant. Thus, the mCHP can achieve stable and repeatable operation.

Figure 4a, b compare the AC electrical efficiencies and the engine efficiencies at lean and stochiometric operation, respectively. The cases run natural gas, the composition of which is detailed in Supplemental Note 5. The results indicate that the lean modes achieve >35% energy efficiency improvement. The maximum AC electrical efficiency and the engine efficiency of the lean combustion are 35.2% and nearly 40%, respectively, whereas the maximum AC electrical efficiency and the engine efficiency of the stoichiometric combustion are 26.4% and 29.2%, respectively (Supplemental Note 6). The superior engine efficiency under the lean modes is further confirmed by Supplemental Note 7, which shows that the exhaust temperature at the lean operations is significantly lower than that of the stoichiometric modes. This indicates that the larger amount of energy released during lean combustion was converted to useful mechanical work rather than into

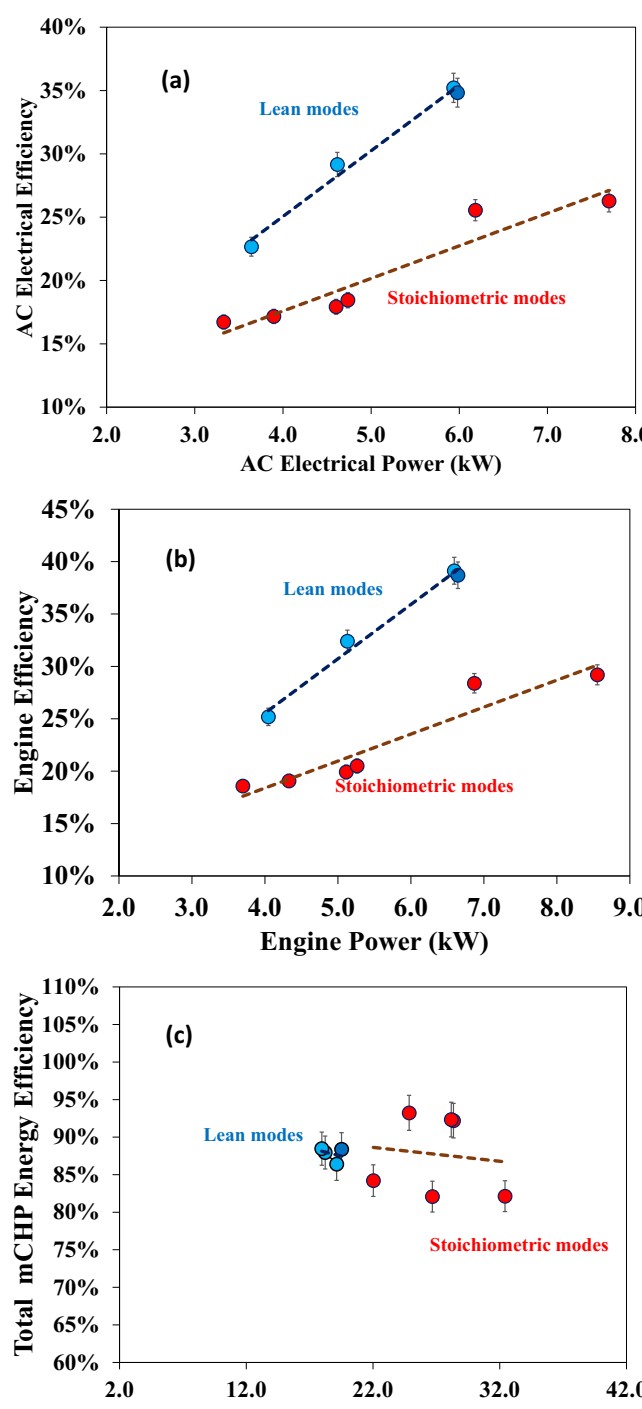

**Fig. 4 | Comparison of mCHP performance between lean and stoichiometric modes. a** Shows AC electrical efficiencies. **b** Shows engine efficiencies. **c** Exhibits overall mCHP efficiencies. The average engine speed is 2864 rpm. Note: the error bars show the measurement uncertainties of electrical, thermal, and total efficiency are 3.28%, 2.26%, and 2.51%, respectively.

exhaust heat, thereby improving the mCHP electrical efficiency. The performance of the OP4S engine was compared with a commercial natural gas engine. Supplemental Note 8 compares engine brake energy efficiency between the OP4S and Cummins Westport natural gas engine as a function of engine power output. Clearly, the OP4S achieves better efficiency at the same power output range. Typically, a comparable natural gas engine tends to have lower efficiency

compared to a gasoline engine due to the liquid nature of gasoline, which allows for better volumetric efficiency than that of a gaseous-fuelled engine. However, when compared to commercial four-stroke port fuel-injected and direct-injection gasoline engines achieving efficiencies around 17% and 22%, respectively, at <8 kW[47–49], the OP4S outperforms, showcasing enhanced engine efficiency. The developed OP4S engine has significantly elevated efficiency, indicating substantial improvement in energy efficiency for mCHP applications.

Figure 4c illustrates the overall mCHP efficiencies operating in lean and stoichiometric combustion modes. For the transient electricity and thermal energy performance of the lean and stoichiometric modes, please see Supplemental Notes 9–12. The observations from Fig. 4c show that the overall mCHP efficiencies in lean combustion modes range from 86.4% to 88.5%. However, the total mCHP efficiency at certain stoichiometric combustion modes can reach >93%. In the stoichiometric case with over 93% efficiency, the exhaust temperature at the exit of the water tank decreases to 31.3 °C. Two factors result in lower overall mCHP efficiencies in the lean combustion modes. First, lean combustion modes result in exhaust temperatures at the water tank exit ranging from 43.4 °C to 47.7 °C, indicating there is potential improvement space in the thermal energy control system to enhance waste heat recovery; second, lean combustion modes inherently lead to a lower ratio of moisture to dry air, increasing uncondensed water in the exhaust flow rejected to ambient conditions and causing higher latent heat loss (see Supplemental Note 13). The results indicate that switching between lean and stoichiometric combustion modes in the mCHP can be optimal to meet flexible thermal energy and electricity demand.

Additionally, emissions such as CO, HC, and $NO_x$ were also measured for all the lean and stoichiometric modes. Detailed results are shown in Supplemental Note 14, which confirms that the prototype using natural gas meets US Environmental Protection Agency new source performance standards (NSPSs) for emissions for SI stationary engines used in the power generation of <19 kW. For engine displacement of the mCHP, the NSPSs require CO emissions <610 g/kWh and HC+$NO_x$ emissions <8 g/kWh[50]. Compared with the stoichiometric modes, all the lean modes yielded substantially lower CO, HC, and $NO_x$ emissions.

Figure 5 shows the examples of thermal energy and exergy flows for mCHP operated under lean and stoichiometric combustion modes in delivering an AC electrical power of around 4.5 kW. Waste heat recovery from the exhaust and coolant flow of the OP4S is comparable in both the stoichiometric and lean modes. The greatest energy loss occurs in the exhaust stream (to ambient conditions), highlighting the importance of exhaust temperature at the exit of the water tank by optimizing the design of the exhaust coil embedded in the water tank. It should be noted that the generator loss is ~1.9%–2.9% of fuel energy, which cannot be recovered. This indicates the importance of adopting a higher efficiency generator, rather than the 90% efficiency generator used in the mCHP.

The exergy flows shown in Fig. 5 reveal that, as expected, the irreversible combustion and heat loss from the combustion chamber to coolant and oil cause the largest exergy loss. Exergy loss owing to the irreversible combustion of the lean and stoichiometric modes is 44.8% and 56.8% of fuel exergy, respectively. Consequently, the lean combustion mode can improve the OP4S power and electricity efficiencies, as shown in Fig. 4(a). The second major exergy loss is heat transfer from high-temperature exhaust flow to water in the tank, an exergy loss of around 16% in both the lean and stoichiometric modes. Thus, a small and compact organic Rankine cycle (ORC) device could be a potential option in the future to recover the exergy loss and boost fuel-to-electricity efficiency. Unlike waste heat recovery from hot exhaust and coolant flow from the OP4S, exergy in coolant flow is limited owing to the stream temperature of 60 °C to ~80 °C, compared

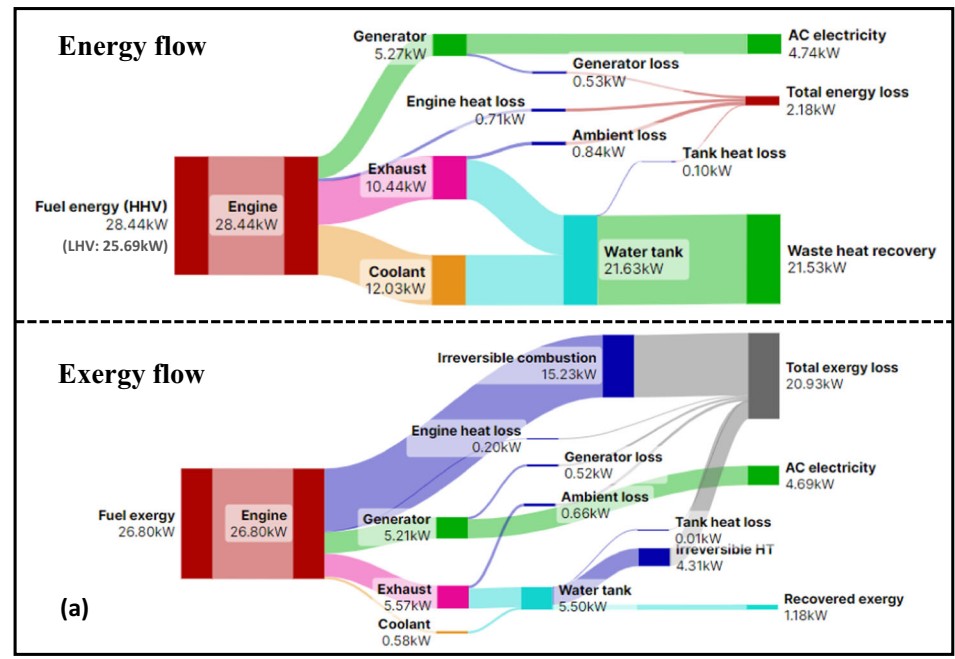

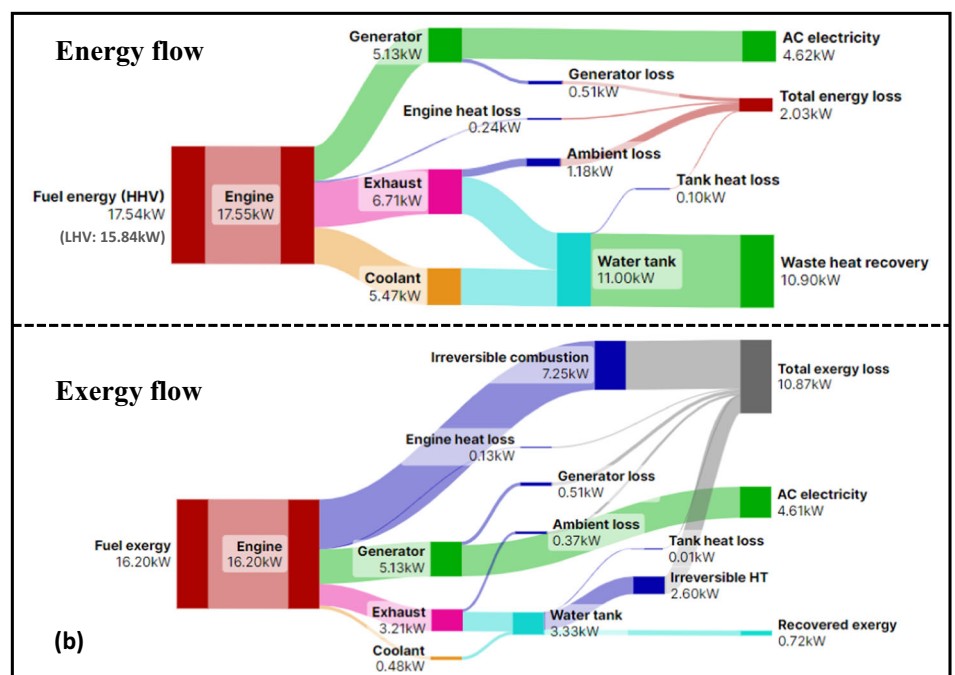

**Fig. 5 | Energy flow and efficiencies for mCHP component and system. a** Presents the stoichiometric combustion mode at 4.74 kW AC (Case 5). **b** Presents the lean combustion mode at 4.62 kW AC (Case 8). Note: exergy loss for irreversible combustion shown in (**a**) includes exergy loss for heat loss from the combustion chamber to coolant and oil.

with exhaust flow. More results on exergy analysis are provided in Supplemental Note 15, where Table S12 shows exergy electrical efficiencies and total mCHP exergy efficiencies. Unlike exergy electrical efficiencies, the total mCHP exergy efficiencies account for very limited water tank exergy recovered besides power generation.

The mCHP has been tested for >600 h without damage or deterioration, indicating its high reliability and safety. The unit has the capability to replace a residential furnace, a water heater, and grid power supply. The unit is recommended for annual replacement of oil and oil filter, spark plug, and air filter, as well as lubrication of the water pump. The detailed cost is shown in Supplemental Note 18, Table S16.

## Discussion
### Discussion of mCHP efficiency

The prototype mCHP system's performance was compared with that of commercial or prototype mCHPs that generate <10 kW of electrical power. The results are shown in Fig. 6, and the fuel type of each model displayed in the figure is shown in Supplemental Note 16. The AC electrical efficiencies are evaluated based on fuel lower heating value (LHV), and the overall mCHP efficiencies are evaluated based on fuel higher heating value (HHV) because a portion of the latent energy in water vapour is usually recovered. The observation reveals that up to 35.2% AC electrical efficiency demonstrated by the developed mCHP operating at lean modes exceeds the upper boundary of 30% for ICE-

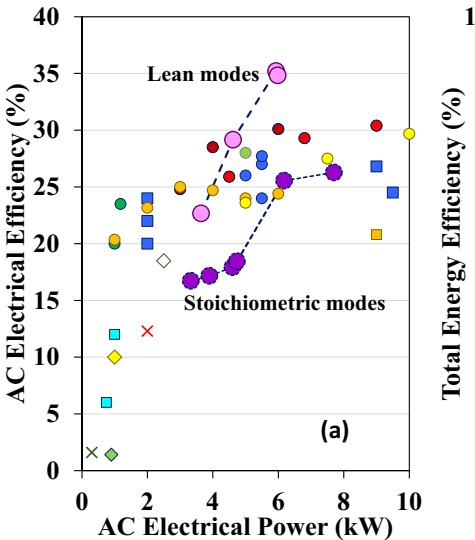
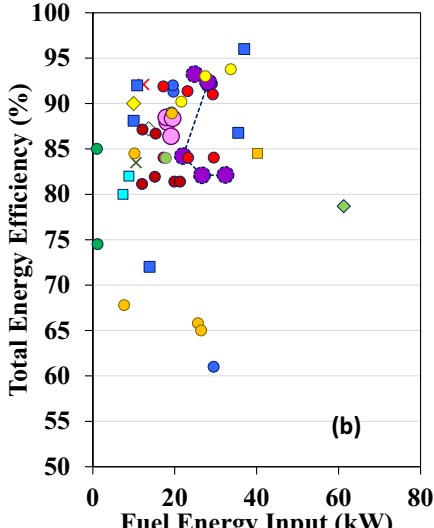

**Fig. 6 | Comparison between the current mCHP and mCHPs reported in the public domain. a** Shows AC electrical efficiencies. **b** Reveals overall mCHP efficiencies. Legend – pink and purple circles: current mCHP at lean and stoichiometric modes, respectively; diamond: Cogen Microsystems(ORC)[3]; yellow diamond: Energetic Genlex (ORC)[3]; red cross: JX Crystal Prototype (TPV)[3]; blue circle: Senertec Daschs (ICE)[3,13,21,23]; blue square: Solo 161 (Stirling)[3,13,22,23]; green circle:

Honda Encowill (ICE)[3,19]; light blue square: Whisper Tech (Stirling)[13,20]; green diamond: Academic prototype (ORC)[16]; green cross: Academic prototype (TPV)[17]; yellow square: Stirling Denmark SM5A (Stirling)[23]; brown circle: EC Power XRGI@6 (ICE)[25]; red circle: EC Power XRGI@9 (ICE)[25]; orange circle: AEG Ecopower (ICE)[26]; yellow circle: Totem Totem 10 (ICE)[27]; light green circle: YANMAR CP5WN (ICE)[28]. All the mCHPs generate electrical power at no more than 10 kW.

based mCHP reported in the public domain. The lean combustion mode is expected to achieve even higher AC electrical efficiency in the proposed mCHP in the cases beyond 6.0 kW. However, hardware limitations precluded running the cases of >6.0 kW under lean combustion modes because the maximum airflow of the engine is limited with wide-open throttle operation. In the studied lean combustion modes, running lean burn with 30% excess air requires reducing the amount of fuel being injected by 30%, thereby limiting the amount of mechanical power the engine can produce at rated power conditions. On the other hand, the overall mCHP efficiency of up to 93.2% at stoichiometric mode is comparable to the maximum mCHP efficiencies reported in the public domain.

The mCHP powered by OP4S technology has significant potential for further improvements, particularly achieving 40% or more AC electrical efficiency. In the current prototype design, only a compression ratio of 8.5 and a premixed lean combustion strategy in the OP4S engine were adopted. The 40% AC electrical efficiency target could be achieved through further approaches, including (1) a higher compression ratio (e.g., 16 or above) with an onset of knock suppressed by lean burn combustion; (2) an even leaner combustion of an equivalent ratio of 0.6, which could be achieved by a spark-assisted, stratified-charge jet ignition system with a prechamber; (3) an implementation of Miller cycle application with a higher expansion ratio (e.g., 20 or above) compared with the compression ratio, improving thermal efficiency; (4) a more efficient generator with 95% efficiency (the current generator efficiency is just 90%); and (5) a compact ORC device used to recover the exergy loss in the OP4S exhaust flow and further boost the fuel-to-electricity efficiency. In addition, if $H_2$ or its mixture with natural gas is used, then the features of $H_2$—such as clean combustion, fast flame speed, and ultralow flammability limit—will allow the mCHP and OP4S engine to operate under a very lean combustion mode with $\lambda > 2.0$, especially with a higher compression ratio. This approach will enable clean and efficient mCHP performance targeting 40% AC electrical efficiency. These endeavours will require adding new components and substantially updating existing hardware and control modules. Consequently, appropriately renovating the mCHP system and enabling better efficiency without significant cost penalty is important for strengthening the practical applications of mCHP

technology and real decarbonization implementation. These approaches deserve further investment and investigation in the future.

## Discussion of renewable fuel effects on mCHP system

The OP4S engine used in the mCHP is optimally designed to operate on natural gas, a reliable and cost-effective energy source with a well-established infrastructure. The OP4S engine is also compatible to run renewable biogas and hydrogen as fuel. However, the optimal operation of renewable biogas and hydrogen in the OP4S engine requires appropriate modifications. The major modifications will focus on the fuel system (i.e., fuel regulators, injectors, pumps, and appropriate pipelines) and the air/fuel mixing system to have appropriate changes in fuel flow rate and air/fuel ratio due to fuel composition variation and in engine ECU recalibration to meet power demand and emissions regulations. The modifications will affect the mCHP cost and efficiency because of different fuel composition and/or energy density compared with natural gas.

Renewable biogas primarily comprises methane with diluents such as nitrogen and carbon dioxide and is considered to enable near-zero greenhouse gas emissions because it is produced from biomass. The efficiencies of the OP4S engines operated on biogas are expected to be comparable to natural gas operation especially at lean operation[51]. One of the key disadvantages of biogas is its inconsistent composition, which may fluctuate with the season and the specific biogas production process. Biogas may also carry impurities and contaminants, including hydrogen sulfide ($H_2S$), siloxanes, ammonia, and moisture. These elements have the potential to erode engine components, foul spark plugs, and damage exhaust systems, ultimately leading to increased maintenance and repair costs. Consequently, the customized engine design is required for managing variability in composition, impurities, and contaminants to accomplish optimal OP4S engine efficiency. Additionally, biogas typically has a lower energy density compared with natural gas (i.e., biogas' calorific value of 20–26 $MJ/m^3$ compared with natural gas' caloric value of 39 $MJ/m^3$), which can result in certain cost penalties because of the need for large-size engines to deliver the given power.

Hydrogen is a fully carbon-free fuel. Compared with natural gas, the high flammability range of hydrogen[52] allows ultra-lean combustion

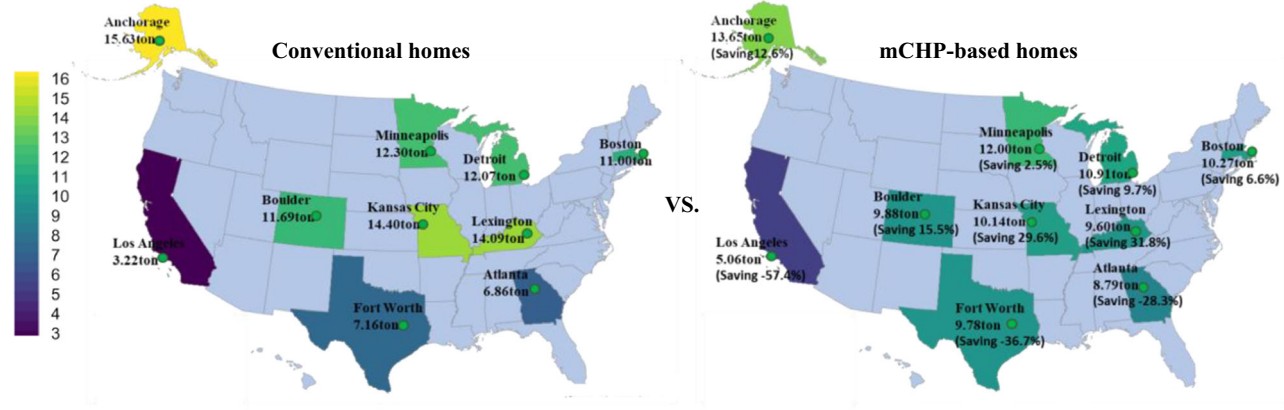

(a) $CO_2$ emissions tons per year

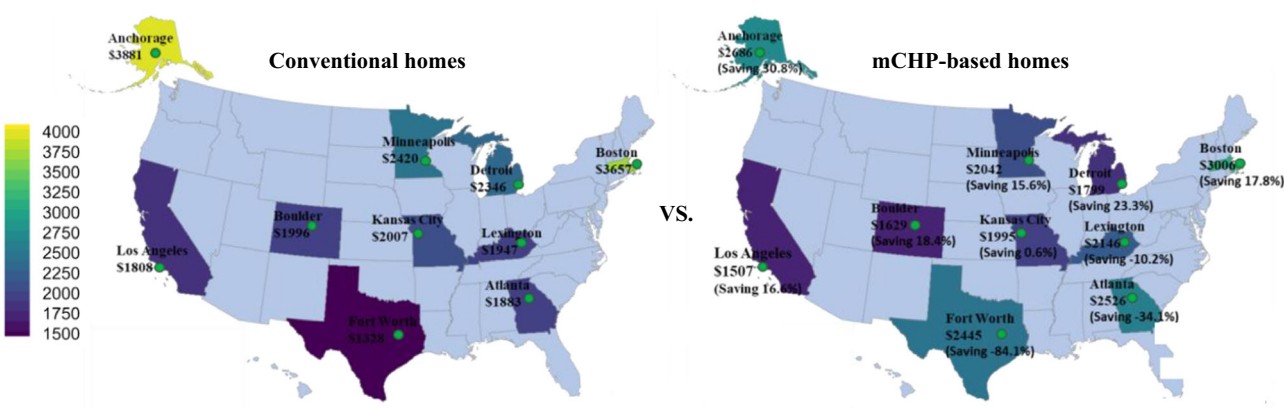

(b) Annual operation cost in US dollars per year

**Fig. 7 | Effect of mCHP applications on annual decarbonization and cost savings of the 10 households.** The households are located in Anchorage (Alaska), Boston (Massachusetts), Detroit (Michigan), Minneapolis (Minnesota), Boulder (Colorado), Kansas City (Missouri), Lexington (Kentucky), Atlanta (Georgia), Fort Worth (Texas), and Los Angeles (California). **a** Shows $CO_2$ emissions tons and decarbonization per year. **b** Exhibits annual operation cost and cost saving in US dollars per year. The colours marked in the figure show annual $CO_2$ emissions and operation cost levels of the selected homes.

in the OP4S, resulting in improved efficiency, lower $NO_x$ emissions, and near-zero carbon emissions into the environment despite a small amount of $CO_2$ emitted owing to the burning of lubrication oil. The hydrogen operation in the OP4S can improve the fuel-to-electricity efficiency by 12.2%–18.8%, based on previous studies[41]. The potential efficiency of OP4S engines using $H_2$ could reach >45% with a peak of about 50%[53]. Like biogas, the drawback of ultra-lean $H_2$ operation also requires a bigger engine to deliver a given power, increasing the mCHP's cost. The high price of hydrogen is caused by insufficient infrastructure, and limited market penetration[54] is another major issue of $H_2$ application in the mCHPs. This indicates substantial need for the development and deployment of $H_2$ production infrastructure, distribution networks, and storage technologies. However, the cost of hydrogen fuel is expected to be competitive with natural gas in the 2035–2050 time frame[52]. This expectation is rooted in the rapid advancement of technology, shifts in energy policies, and evolving global market dynamics[55].

Therefore, the proposed mCHPs powered by the OP4S engine not only can be employed today to benefit regions with highly polluting electrical grids but also can serve as a promising foundation for the transition from conventional fossil fuels to zero carbon emissions in the future.

**Discussion of the cost savings and decarbonization of mCHP**
US electricity in 2023 was produced from wide-ranging sources such as coal (14.9%), hydroelectric (6.2%), natural gas (40.1%), nuclear

power (19.6%), solar (4.4%), wind energy (13.5%), and others (2.3%). Fossil fuels such as coal and natural gas power are still dominant, but renewable sources such as wind and solar are growing quickly in several states. Consequently, carbon intensity of electricity generation (i.e., kilograms of $CO_2$ per kilowatt-hour of power generation) and electricity retail price vary substantially at different locations. To reasonably evaluate the benefits and disadvantages of the mCHP prototype performance in the decarbonization and operation cost savings of single household applications, 10 homes were selected from cities in 10 states, respectively, representing northern, middle, and southern climate zones in the United States (see Supplemental Note 17). Anchorage (Alaska), Boston (Massachusetts), Detroit (Michigan), and Minneapolis (Minnesota) were selected to represent the northern climate zone. Boulder (Colorado), Kansas City (Missouri), and Lexington (Kentucky) were selected to represent the middle climate zone. Atlanta (Georgia), Fort Worth (Texas), and Los Angeles (California) were selected to represent the southern climate zone. The study employed a strategy to optimally use the electricity and heat from the mCHP for a single household application. This method, as well as the calculation method for the evaluation of emissions savings and cost reductions, is detailed in Supplemental Note 18. The results are shown in Supplemental Note 19. Figures 7 and 8 summarize the potential decarbonization and cost savings. In the study, the fuel used in the mCHP is natural gas. The detailed decarbonization and cost savings vary with these locations' carbon intensity of electricity generation, electricity and natural gas price, and climate conditions

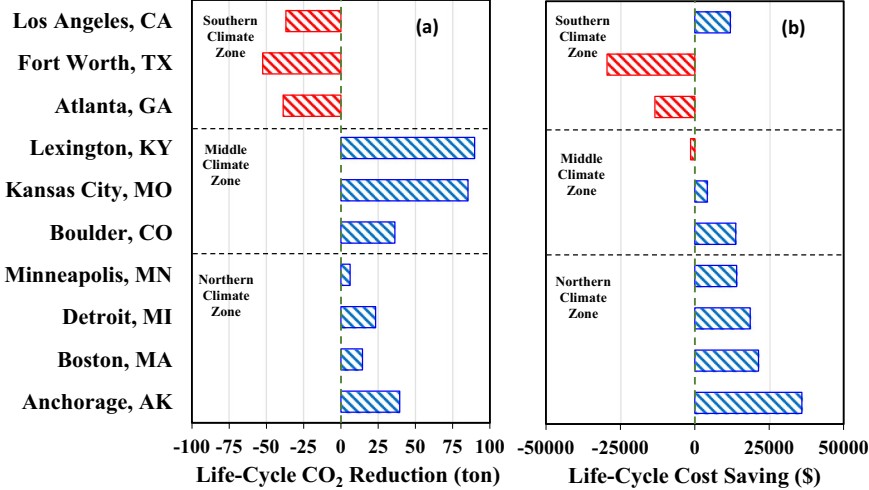

**Fig. 8 | Comparison of life cycle decarbonization and cost savings by mCHP applications. a** Shows life cycle decarbonization. **b** Exhibits cost savings. The life span of mCHP is 20 years. Red bars indicate a penalty associated with $CO_2$ reduction or cost when deploying mCHP, while blue bars indicate a benefit associated with $CO_2$ reduction or cost when deploying mCHP.

enabling the optimal utilization of electricity and waste heat from the mCHP.

Figure 7 shows that the mCHP enables annual decarbonization and operation cost savings. It achieves 2.5%–12.60% $CO_2$ reduction (i.e., 0.31–1.96 tons per year) and 15.6%–30.8% cost savings (i.e., $378–$1,194 per year) in the northern climate zone. In the middle climate zone, the mCHP enables 15.5%–31.8% $CO_2$ reduction (i.e., 1.81–4.49 tons per year) but results in a mixed scenario of −10.2%–18.4% cost savings (i.e., −$198–$367 per year). The reason for significant $CO_2$ reduction in the middle zone is that most power plants in these locations are coal-fired, leading to higher carbon intensity of electricity generation (see Fig. S11 at Supplemental Note 17). However, in the southern climate zone, the decarbonization diminishes owing to lower thermal demand, which is a result of a shorter cold season requiring less space heating (see Fig. S13 at Supplemental Note 17). This, in turn, results in a significant operational cost penalty for the southern regions considered except Los Angeles, California. Los Angeles has an electricity price 81.2%–115.3% higher than that of the other two locations in the southern climate zone (see Fig. S11 at Supplemental Note 17), thus yielding better economic benefits. Figure 8 illustrates the life cycle decarbonization and cost savings of mCHP applications in the selected 10 households. The observation of Fig. 8a shows that the life cycle decarbonization could be up to 39.48 tons of $CO_2$ in the northern climate zone and up to 89.77 tons of $CO_2$ in the middle zone, but the decarbonization potential in the southern zone is negative, as confirmed by the results shown in Fig. 7a. Figure 8b reveals that the mCHP achieves substantial life cycle cost savings in the northern climate zone owing to its long life span and substantial waste heat recovery (or less waste heat loss). The long service life of the mCHP saves initial investment and enables no intermediate replacement cost compared with those of furnaces and water heaters (see Supplemental Note 18).

Therefore, except for specific locations, the mCHP excels in achieving decarbonization and cost savings primarily in the northern and central climate zones. On the other hand, even in the southern climate zone, the mCHP can still offer a solid solution that satisfies the breadth of both electrical and thermal energy needs in remote and underserved communities due to their geographic isolation and high-potential risk of energy disruptions and natural disasters[56]. Overall, the mCHP prototype is inherently suited to cold climate regions but also can play a critical role for the energy needs of remote and underserved communities in all climate zones.

## Results and discussion

A compact and portable mCHP prototype at TRL 6 has been developed with several features, including adoption of the highly efficient OP4S engine, a flexible fuel capability, and a well-designed waste heat recovery system capable of recovering two waste heat sources (i.e., exhaust and hot coolant) to provide hot water in meeting thermal load demand. The mCHP prototype enables up to 35.2% of fuel-to-electricity efficiency and nearly 93% of the overall mCHP efficiencies, exceeding the conventional mCHP's fuel-to-electricity efficiency limit of 30%. Achieving high electrical efficiency is always critical and challenging in the development of new mCHP technologies. Moreover, the OP4S engine has 60% fewer parts per engine unit, which, therefore, lowers material and manufacturing costs while enabling longer service time. The combination of high efficiency and cost-effectiveness enables significant potential deployment and market penetration for the mCHP in the US residential sector. In addition, the mCHP prototype can be powered by traditional fuels such as natural gas or propane but also has the potential to run on carbon-free fuels such as hydrogen. These features indicate the technology has a substantial potential of supporting the transition from current conventional fossil fuels to carbon-free fuels in the future. In addition, the mCHP is a compact and portable device, allowing high versatility for installation locations.

The mCHP emissions, including CO, HC, and $NO_x$, were also measured from all the lean and stoichiometric modes. The results meet US NSPS emissions for SI stationary engines used in the power generation of <19 kW. Compared with the stoichiometric modes, all the lean modes have substantially lower CO, HC, and $NO_x$ emissions. The decarbonization and cost savings potential of mCHP were studied. The detailed decarbonization and cost savings vary with the locations' carbon intensity of electricity generation, electricity and natural gas price, and climate conditions. Except for specific locations, the mCHP excels in achieving decarbonization and cost savings primarily in the northern and middle climate zones. In the southern climate zone, decarbonization is less effective owing to the shorter cold season, which results in less waste heat recovery for space heating and significant cost penalties.

Overall, the mCHP prototype is inherently suited to cold climate regions but also can play a critical role for the energy needs of remote and underserved communities in all climate zones. The remote and underserved communities usually do not have sufficient grid and energy infrastructure, especially during unexpected severe weather when both electricity and heat are especially needed. The developed

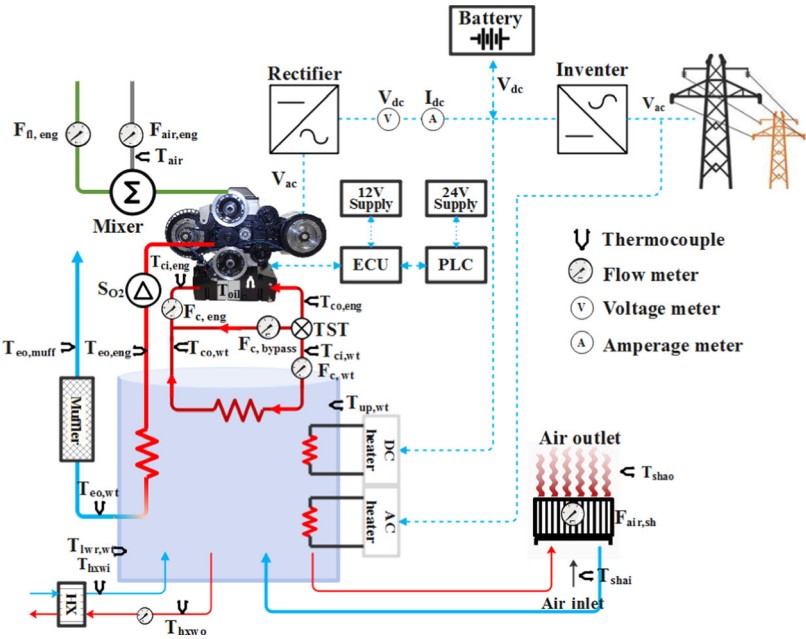

| Label | Description | Label | Description |
|---|---|---|---|
| $T_{eo,eng}$ | Thermocouple for exhaust at engine out | $F_{fl,eng}$ | Natural gas flow for engine |
| $T_{eo,wt}$ | Thermocouple for exhaust at water tank out | $F_{air,eng}$ | Airflow for engine |
| $T_{eo,muff}$ | Thermocouple for exhaust at muffler out | $F_{c,eng}$ | Coolant flow for engine |
| $T_{co,eng}$ | Thermocouple for coolant at engine out | $F_{c,wt}$ | Coolant flow for water tank |
| $T_{ci,eng}$ | Thermocouple for coolant at engine in | $F_{c,pass}$ | Coolant flow for bypass loop |
| $T_{oil}$ | Thermocouple for engine oil | $F_{air,sh}$ | Airflow for space heating |
| $T_{air}$ | Thermocouple for engine air in | $F_{hxwf}$ | Heat exchanger water flow |
| $T_{ci,wt}$ | Thermocouple for coolant at water tank in | $S_{O2}$ | Oxygen sensor |
| $T_{co,wt}$ | Thermocouple for coolant at water tank out | TST | Thermostat mechanical flow valve |
| $T_{up,wt}$ | Thermocouple for water at upper tank | $V_{ac}$ | AC voltage |
| $T_{lwr,wt}$ | Thermocouple for water at lower tank | $V_{dc}$ | DC voltage |
| $T_{hxwi}$ | Thermocouple for water at heat exchanger in | $I_{dc}$ | DC current |
| $T_{hxwo}$ | Thermocouple for water at heat exchanger out | ECU | Engine control unit |
| $T_{shai}$ | Thermocouple at space heating air in | PLC | Programmable logic controller |
| $T_{shao}$ | Thermocouple at space heating air out | Mixer | Air and fuel mixer |

**Fig. 9 | Schematic of the mCHP testing system and sensor installation map.** Blue dashline represents electrical energy; red solid line represents hot thermal energy; blue solid line represents cold thermal energy; and gree solid line represents fuel energy. Note: ECU is engine control unit; PLC is programmable logic controller; and AC and DC are alternating and direct current, respectively.

mCHP is expected to promote acceptance and accelerate the adoption of mCHP for residential and light commercial building markets in the communities mentioned above. The proposed mCHP is expected to enter the commercial market within 3–5 years.

## Methods
### mCHP experimental setup
The developed mCHP was installed and tested in a test cell at Enginuity's testing facility in Clinton Township, Michigan, and pictured in Fig. 9. Because the testing facility and real residential homes have different infrastructures for hot water supply and space heating applications, a thermal load was alternatively imposed by adding an intermediate compact plate heat exchanger to transfer heat from hot water in the mCHP's water tank to the cooling loop in the testing facility. The cooled supply water was then returned to the water tank instead of to the real direct hot water supply. The space heating application was designed to directly deliver hot water from the mCHP's water tank to a space heating device.

To collect data for comprehensively analysing the mCHP system, the thermocouples and flow metres for air, fuel, and coolant, as well as the sensors for electricity current and voltage measurement, were installed in the system. The table listed in Fig. 9 further explains the detailed measurement parameters for flows, temperatures, and electricity. The key measurements include fuel consumption, battery current and voltage, lambda sensor or $O_2$ sensor, exhaust temperature at the inlet and outlet of the water tank, coolant temperature at the inlet and outlet of the water tank, coolant flow, water flow and inlet and outlet temperature for household hot water supply, and water flow and inlet and outlet temperature for space heating. The data from these sensors are collected by the PLC modules, which, along with the ECU, control the mCHP systems.

### Efficiency analysis methodology
In the study, the electrical efficiency, $\eta_e$, is defined by the first law of thermodynamics as electrical power, $\dot{w}_e$, divided by the fuel energy consumption rate, $\dot{Q}_{f,LHV}$, based on the LHV. The electrical energy $\dot{w}_e$

considered here could be AC electricity produced by the generator or DC electricity produced by the rectifier. The AC electrical efficiency, $\eta_{e,ac}$, and DC electrical efficiency, $\eta_{e,dc}$, can also be described as a function of engine efficiency, generator efficiency, and rectifier efficiency, given by Eqs. (1) and (2).

$$\eta_{e,ac} = \frac{\dot{w}_{e,AC}}{\dot{Q}_{f,LHV}} = \eta_{eng} \cdot \eta_{gen} \tag{1}$$

$$\eta_{e,dc} = \frac{\dot{w}_{e,DC}}{\dot{Q}_{f,LHV}} = \eta_{eng} \cdot \eta_{gen} \cdot \eta_{rect} \tag{2}$$

An mCHP typically produces useful thermal energy in addition to electricity. Consequently, the overall mCHP efficiency, $\eta_{chp}$, is addressed by adding the useful thermal energy flow, $\dot{Q}_{th}$, to the electrical energy rate, $\dot{w}_e$, and dividing by fuel energy consumption flow, $\dot{Q}_{f,HHV}$, based on fuel HHV. In the current CHP configuration, $\eta_{chp}$ is shown in Eq. (3) based on the AC electrical energy rate.

$$\eta_{chp} = \frac{\dot{w}_{e,AC} + \sum \dot{Q}_{th}}{\dot{Q}_{f,HHV}} \tag{3}$$

In these equations, the fuel energy consumption flows, $\dot{Q}_{f,HHV}$ and $\dot{Q}_{f,LHV}$, are calculated based on fuel mass flow multiplied with fuel HHV and LHV, respectively. The HHV (also known as the gross calorific value) of a fuel is defined as the amount of heat released by a specified quantity (initially 25 °C) once it is combusted and the products have cooled to a temperature of 25 °C, which considers the latent heat of water vapour in the combustion products. The LHV (also known as the net calorific value) of a fuel is defined as the amount of heat released by combusting a specified quantity of fuel (initially at 25 °C) and cooling the temperature of the combustion products to 150 °C; in this process, the latent heat of vaporization of water in the reaction products is not recovered.

**Methodology of exergy analysis**
By assuming a steady-state condition ignoring kinetic and potential energy, the exergy rate equations are developed for each component in the mCHP (i.e., engine, generator, waste heat recovery component, and other components). A thorough exergy analysis of the engine component was conducted to account for the exergy associated with fuel, work, exhaust gas, engine coolant, engine heat loss, and mechanical work. The exergy destroyed during the irreversible combustion process and heat loss from the combustion chamber to coolant and oil is determined by contrasting the exergy of the fuel with the residual exergy mentioned above. In addition, the condensation of water in exhaust gas exiting the waste heat recovery system was considered based on the saturation pressure at exhaust gas temperature exiting the waste heat recovery system. The details are addressed in Supplemental Note 15.

**Methodology of economic analysis and decarbonization.** To assess the potential economic and environmental benefits of the proposed mCHP in building applications, a comprehensive analysis of life cycle cost savings and carbon emissions has been conducted for 10 representative residential houses (see Supplemental Note 17). In the economic analysis, the mCHP is assumed to replace residential furnace, water heater, and grid power supply. The life cycle analysis for the mCHP, established over 20 years, accounts for annual operation and maintenance cost savings, initial investment penalty, and disposal cost penalty at the end of its life cycle. Annual operation cost savings are determined by the operation cost of mCHP minus the combined operation cost of a residential furnace, water heater, and grid power supply in each home. Because the mCHP enables flexible electricity

outputs to meet dynamic electricity and thermal energy demands, the mCHP operates optimally by switching between stoichiometric and lean modes, considering a trade-off of cost savings and carbon emission reduction while meeting thermal energy demand and electricity demand in a home. In each mCHP operation mode, the electricity output is used to satisfy household power demand. However, if the mCHP electricity output falls short of meeting household power demand, grid electricity buffers the excess power demand. Similarly, the mCHP waste heat is used to satisfy household thermal energy demand for space and water heating typically fulfilled by natural gas. However, if the mCHP waste heat is insufficient to satisfy thermal energy demand, the mCHP uses its electricity output, along with grid electricity, to buffer the additional thermal energy demand. This operating strategy aims to maximize utilization of the mCHP energy output. Annual maintenance cost savings, initial investment penalty, and disposal cost penalty are assessed based on the cost differentials between the mCHP and the residential device it replaces. The life cycle carbon emissions from the mCHP have also been performed based on accounting for direct carbon emissions associated with natural gas consumption in the mCHP and indirect carbon emissions imported from the grid. In addition, as a benchmark, the carbon emissions from a conventional house, which uses electrical components and natural gas devices (e.g., gas furnace and water heater), were also calculated to evaluate the potential benefits of the mCHP. This comprehensive analysis provides insights into both the economic viability and environmental benefits of integrating the proposed mCHP system in residential settings. The details are described in Supplemental Notes 18, 19 and 20.

## Data availability
The experimental and numerical data generated in this publication is available as Source Data file and have also been deposited in the figshare database with access link of https://figshare.com/articles/dataset/Source_Data_xlsx/25596231. Source data are provided with this paper.

## Code availability
The data analysis codes that support the findings of the study are available from the corresponding author upon request.

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

## Acknowledgements

The authors would like to acknowledge U.S. Department of Energy, Office of Energy Efficiency & Renewable Energy, The Building Technologies Office (Funding No. TCF-16-12190). Special thanks go to Jinsu Kim for the help of exergy analysis and Rachel Brooks and Wendy Hames for technical editing. Notice: This manuscript has been authored by UT-Battelle LLC under contract DE-AC05-00OR22725 with the US Department of Energy (DOE). The US government retains and the publisher, by accepting the article for publication, acknowledges that the US government retains a nonexclusive, paid-up, irrevocable, worldwide license to publish or reproduce the published form of this manuscript, or allow others to do so, for US government purposes. DOE will provide public access to these results of federally sponsored research in accordance with the DOE Public Access Plan (http://energy.gov/downloads/doe-public-access-plan).

## Author contributions

Z.G., P.Z. and J.B.L. supervised the project. Z.G., P.Z. and J.B.L. conceived the idea. Z.G., P.Z., J.B.L., T.M. and J.M. designed and Performed experiments. Z.G., P.Z., J.B.L., J.M. and M.M. analyzed data. Z.G., J.M., M.M., M.Z., P.C., A.A., H.L. and K.N. contributed materials and provided resources. Z.G., P.Z., and J.B.L. wrote the manuscript. Z.G., P.Z., J.B.L., T.M., M.M., H.L., B.F. and K.N. revised the manuscript. All the authors reviewed and agreed on the final manuscript.

## Competing interests

The authors declare no competing interests.
