## [Peer Review File · Nature Communications]

Development of a micro-combined heat and power powered by an opposed-piston engine in building applicationsREVIEWER COMMENTS

Reviewer #1 (Remarks to the Author):

The authors developed a prototype of a new mCHP based on a small capacity opposed-piston four-stroke (OP4S) engine for the combined provision of electricity and heat to end-users, especially for single-family houses or light commercial buildings. I appreciate the manuscript, the authors made good work and they showed an important potential application of the OP4S engine in micro cogeneration applications. However, the work needs some improvement (revision) and my suggestion for improvement are as follows:

- The technical (e.g., efficiency) and economic effect of the different types of fuels in the developed mCHP are not well addressed, except for mentioning that the OP4S engine can use fuels like renewable biogas, natural gas, propane, or hydrogen.
- What are the advantages and disadvantages of the different types of fuel used in the mCHP?
- To account for the quality of the mCHP outputs (electricity and heat), an exergy analysis of the system may be interesting to be carried out.

Reviewer #2 (Remarks to the Author):

This paper shows performance of a prototype mCHP system for residential buildings. The result are impressive. There are some comments below:

- (1)As mentioned in the paper, the mCHP system enables flexible electricity outputs to meet dynamic electricity and thermal energy demands. So more attention should be paid on the management of different energy flows to make them match well with building's energy demands. Please specify the control strategies of different modes for the mCHP prototype.
- (2)The results are impressive and the work support the conclusions well. But the advance of the system lies in the advanced OP4S engine rather than the whole mCHP system. The concept of mCHP systems based on waste heat recovery systems and energy storage systems has been widely reported. Please make a more clear novelty statement of the work.
- (3)It is recommended to use a Sankey diagram to represent Figure 5, making the flow and distribution of energy more clear and comprehensible.

Reviewer #3 (Remarks to the Author):

General comments:

1. The motivation for the proposed mCHP technology deployment in residential/"light commercial" buildings should be elaborated upon in the context of/relative to other competing low-carbon technologies, including renewable electricity and renewable heat supply. In addition to thermodynamic efficiency, the fuel used in the proposed mCHP technology, and its expected penetration in buildings, will determine potential emissions abatement globally in a given timeline. "Light commercial" buildings needs to be defined.
2. Among the potential fuels listed (i.e., biogas, natural gas, propane or hydrogen), are natural gas and propane effective/viable fuel options considering global decarbonization goals – what is the anticipated timeline for the deployment of the proposed mCHP technology. The emission levels achievable with different fuels would need to be quantified, including upstream emissions in the case of fossil gas. As part of "performance" metrics, emissions other than GHG also need to be quantified. The fuel employed to generate the experimental/simulation results presented, and compare the performance of the proposed system (energy, emissions) with competing technologies/system(s) of reference, needs to be explicitly documented throughout the manuscript including as part of figures/tables of results; Line 106 indicates natural gas; can its composition be documented. How is thermodynamic performance expected to be influenced by fuel, and thus comparison of performance with competing system(s)/system(s) of reference.
3. Experimental measurement uncertainty, tested repeatability, reproducibility require to be further

documented. Experimental uncertainty needs to be visualized in figures/tables of results to make the presentation of the results more self-explanatory.

4. The system flexibility with regard to power to heat ratio to meet the needs of different (building) users would benefit from more explicit documentation of the power-to-heat ratios and their effect on thermodynamic performance. Full load versus part load performance also requires more explicit documentation as part of the results.

5. Analyzing and reporting second-law based thermodynamic performance would provide additional insight into the performance of the proposed mCHP technology in the different test configurations/cases considered, and how it compares with system(s) of reference (e.g., Cummins system, systems in Table S1).

6. CFD simulations results are presented focusing on the water storage tank. CFD model validation (i.e., comparison between CFD and corresponding experimental results) should be presented. The objectives of CFD simulation should be further clarified in the context of the experimentally focused work, i.e. contribution of the simulation results to the overall manuscript analysis.

7. Have experimental results presented in the manuscript been compared with corresponding thermodynamic model predictions, which could be useful to extend the results presented to other test configurations.

8. Maintenance, reliability, lifetime, safety, recyclability and cost are other aspects for application of the proposed mCHP technology to buildings, which could be (further) addressed in the manuscript.

9. The main general comment is concerned with the significance of the manuscript contents for possible publication. If the proposed technology is already commercialized, the potential novelty and advancement contributed by the manuscript submission require to be clarified and further highlighted in the manuscript submission. The analysis methods employed in the manuscript (experimental/modeling-simulation) are not considered novel or state of the art (including based on above comments). The results (Fig. 6) indicate that the proposed system can reach high-bound mCHP efficiencies; what would be globally the emissions saved by the proposed technology in what timeline considering its expected penetration in buildings; would the proposed technology be considered a key technology to achieve net-zero emissions energy systems by mid century.

Specific comments:

- Manuscript submission title: The title does not refer to buildings as the proposed mCHP technology application. Could the manuscript submission title be revised accordingly; otherwise could this omission in the existing title be justified.
- Abstract: The fuel and analysis methods require to be documented in the abstract, as well as measured/modeled GHG emissions savings relative to competing technology(ies)/system(s) of reference. The last sentence in the abstract needs to be supported by evidence presented in the manuscript. The abstract does not refer to economics.
- Lines 33-35: What is the purpose of the statement; for example what geographical area(s) is(are) targeted for application in the manuscript submission. Should specific climate or socio-economic aspects be clarified.
- Lines 54-70: The timeline of development of the proposed mCHP technology and current/future TRL need to be documented.
- Line 72: "small" in "a small OP4S engine" is to be quantified.
- Lines 76-79: The potential suitability of the proposed mCHP technology in buildings needs to be further elaborated based on additional criteria, in the introduction and results/discussion sections of the manuscript submission." Light-commercial buildings" needs to be defined (if possible, quantitatively).
- Line 77: Is "low cost" an outcome of the analysis documented in the submission? If not, it should be quantified/defined with appropriate citations.
- Lines 77-79: The following text requires further elaboration (i.e., drop in, connections, vulnerable energy infrastructure and high energy cost) in "drop-in replacement feature using the building's existing connections, particularly in remote and underserved communities which faces significant challenges due to vulnerable energy infrastructure and high energy cost". Please consider incorporating examples to support this claim.
- Line 115: Please quantify.
- Figure 6: Please specify fuel for all devices considered, and whether competing systems data is directly from manufacturers and the operating condition considered (e.g., full load/design conditions). Are all efficiency definitions used by the different cited sources to quantify device

performance consistent.

- Line 253: How are the emissions savings and cost reductions estimated (methods including assumptions and calculations). In Figure 7, could a legend be included to define colour coding. Including % CO₂ emissions and % cost savings could be valuable. The figure text font graphical quality requires improvements.
- Conclusions, Lines 273-274: The following text "can use a variety of fuels", "can provide flexible and thermal electrical loads", "the OP4S is simpler and cheaper to manufacture, assemble and operate" are not found to be supported by analysis presented in the manuscript. If such information was sourced and demonstrated elsewhere, it would be more appropriately placed in earlier parts of the manuscript (e.g., introduction) with cited references.
- Conclusions, Line 277: "if appropriate technologies can be applied to overcome hardware limitations" is vague. What are the technologies and hardware limitations referred to.
- Conclusions, Line 278: the methods employed for calculating emissions savings need to be elaborated in the manuscript. In general, decarbonization potential should consider not only specific emissions savings but also the expected technology penetration potential in buildings. No economic analysis is found in the manuscript to support "...cost savings potential were studied".

Supplementary information document:

- In order to be more self-explanatory, figures of results (such as Fig. S1) and tables of results should specify the fuel composition used in the proposed mCHP technology.
- In Table S1, what are the intended applications of the competing mCHP systems listed. Are the electricity/heat outputs and electrical/overall energy efficiencies obtained directly from mCHP vendors and in what conditions (e.g., part and full load/design conditions). Is the temperature level of the heat output available. Comments on the main reasons for differences in thermodynamic performance between reported systems would be informative. Are vendor data obtained using specific procedures/standards.
- Table S2/S3: Additional information on controls for water outlet temperature.
- In Tables S6 and S7, including the available waste heat and percent waste heat recovery (relative to available waste heat) may be of interest. The electrical and thermal efficiencies could also be included.
- Concerning the comparison between the proposed mCHP system and system of reference (e.g., Cummins system), for each test configuration (case), in order to provide additional insight, electrical and thermal efficiencies, overall energy efficiencies, as well as second law efficiencies, would require to be compared.

General response

We have diligently revised the manuscript, incorporating feedback from the reviewers. This involved rephrasing text, updating figures (including those in the manuscript and numerous supplemental documents), and addressing the comments put forth by the reviewers. Additionally, we have included relevant references in both the revised manuscript and the supplemental document. If there are any references that the reviewers feel we may have overlooked, we encourage them to provide a list, and we will ensure their inclusion in the subsequent revision. Below, you will find the original reviewer comments, followed by our responses. We believe that these revisions represent a substantial improvement and sincerely appreciate the reviewers' valuable time and comments/suggestions.

Reviewer #1:

1.1 The technical (e.g., efficiency) and economic effect of the different types of fuels in the developed mCHP are not well addressed, except for mentioning that the OP4S engine can use fuels like renewable biogas, natural gas, propane, or hydrogen.

Author response: in Page 11: lines 303–340, we address the question and Q1.2 together as below:

“The OP4S engine used in the mCHP is optimally designed to operate on natural gas, a reliable and cost-effective energy source with a well-established infrastructure. The OP4S engine is also compatible to run renewable biogas and hydrogen as fuel. However, the optimal operation of renewable biogas and hydrogen in the OP4S engine requires appropriate modifications. The major modifications will focus on the fuel system (i.e., fuel regulators, injectors, pumps, and appropriate pipelines) and the air/fuel mixing system to have appropriate changes in fuel flow rate and air/fuel ratio due to fuel composition variation and in engine ECU recalibration to meet power demand and emissions regulations. The modifications will affect the mCHP cost and efficiency because of different fuel composition and/or energy density compared with natural gas.

Renewable biogas primarily comprises methane with diluents such as nitrogen and carbon dioxide and is considered to enable near-zero greenhouse gas emissions because it is produced from biomass. The efficiencies of the OP4S engines operated on biogas are expected to be comparable to natural gas operation especially at lean operation [51]. One of the key disadvantages of biogas is its inconsistent composition, which may fluctuate with the season and the specific biogas production process. Biogas may also carry impurities and contaminants, including hydrogen sulfide (H₂S), siloxanes, ammonia, and moisture. These elements have the potential to erode engine components, foul spark plugs, and damage exhaust systems, ultimately leading to increased maintenance and repair costs. Consequently, the customized engine design is required for managing variability in composition, impurities, and contaminants to accomplish optimal OP4S engine efficiency. Additionally, biogas typically has a lower energy density compared with natural gas (i.e., biogas' calorific value of 20–26 MJ/m³ compared with natural gas' caloric value of 39 MJ/m³), which can result in certain cost penalties because of the need for large-size engines to deliver the given power.

Hydrogen is a fully carbon-free fuel. Compared with natural gas, the high flammability range of hydrogen [52] allows ultra-lean combustion in the OP4S, resulting in improved efficiency, lower NO_x emissions, and near-zero carbon emissions into the environment despite a small amount of CO₂ emitted owing to the burning of lubrication oil. The hydrogen operation in the OP4S can improve the fuel-to-electricity efficiency by 12.2%–18.8%, based on previous studies [41]. The potential efficiency of OP4S engines using H₂ could reach more than 45% with a peak of about 50% [53]. Like biogas, the drawback of ultra-lean H₂ operation also requires a bigger engine to deliver a given power, increasing the mCHP's cost. The high price of hydrogen is caused by insufficient infrastructure, and limited market penetration [54] is another major issue of H₂ application in the mCHPs. This indicates substantial need for the development and deployment of H₂ production infrastructure, distribution networks, and storage technologies. However, the cost of hydrogen fuel is expected to be competitive with natural gas in the 2035–2050 time frame [52]. This

expectation is rooted in the rapid advancement of technology, shifts in energy policies, and evolving global market dynamics [55].

Therefore, the proposed mCHPs powered by the OP4S engine not only can be employed today to benefit regions with highly polluting electrical grids but also can serve as a promising foundation for the transition from conventional fossil fuels to zero carbon emissions in the future.”

1.2 What are the advantages and disadvantages of the different types of fuel used in the mCHP?

Author response: The advantages and disadvantages of the different types of fuel used in the mCHP have been addressed together in Q1.

1.3 To account for the quality of the mCHP outputs (electricity and heat), an exergy analysis of the system may be interesting to be carried out.

Author response: The exergy analysis of the mCHP has been conducted and reported in the revision. A brief introduction on the exergy analysis methodology is added in the Methods; see below. For the detailed exergy analysis methodology, please see Supplemental Note 15. In addition, the sample of the exergy analysis is plotted in Figure 5.

In lines 485–493: “By assuming a steady-state condition ignoring kinetic and potential energy, the exergy rate equations are developed for each component in the mCHP (i.e., engine, generator, waste heat recovery component, and other components). A thorough exergy analysis of the engine component was conducted to account for the exergy associated with fuel, work, exhaust gas, engine coolant, engine heat loss, and mechanical work. The exergy destroyed during the irreversible combustion process and heat loss from the combustion chamber to coolant and oil is determined by contrasting the exergy of the fuel with the residual exergy mentioned above. In addition, the condensation of water in exhaust gas exiting the waste heat recovery system was considered based on the saturation pressure at exhaust gas temperature exiting the waste heat recovery system. The details are addressed in Supplemental Note 15.”

(a)

(b)

Figure 5. Energy flow and efficiencies for mCHP component and system with waste heat recovered and stored in the water tank. (a) Stoichiometric combustion mode at 4.74 kW AC (Case 5); (b) lean combustion mode at 4.62 kW AC (Case 8). Note: exergy loss for irreversible combustion shown in Figure 5(a) includes exergy loss for heat loss from the combustion chamber to coolant and oil.

Reviewer #2:

2.1 As mentioned in the paper, the mCHP system enables flexible electricity outputs to meet dynamic electricity and thermal energy demands. So more attention should be paid on the management of different energy flows to make them match well with building's energy demands. Please specify the control strategies of different modes for the mCHP prototype.

Author response: The mCHP prototype was developed to simultaneously provide electricity and heat to residential or light commercial buildings. We address the above description in lines 100–106: “The electricity and thermal energy outputs in this mCHP system can be adjusted by controlling the fuel flow rate and air-fuel ratio. Appropriate changes in spark timing of the OP4S engine will further affect the ratio of the electricity over thermal energy output available in exhaust gas while achieving reasonable engine efficiency. Moreover, the mCHP prototype is designed to connect the electrical battery module, which can flexibly deliver electricity by switching between various combustion modes to meet dynamic electrical and thermal energy demands of residential and light commercial buildings.”

We also address the control strategies of different modes for the mCHP in Methodology of Economic Analysis and Decarbonization (see lines 441–450 at Supplemental note 18) as “In that strategy, the mCHP operates optimally by switching between stoichiometric and lean modes, considering a trade-off of cost savings and carbon emissions reduction while meeting thermal energy and electricity demands in a home. In each mCHP operation mode, the electricity output is used to satisfy household power demand. However, if the mCHP electricity output falls short of meeting household power demand, grid electricity buffers the excess power demand. Similarly, the mCHP waste heat is used to satisfy household thermal energy demand for space and water heating typically fulfilled by natural gas. However, if the mCHP waste heat is

insufficient to satisfy thermal energy demand, the mCHP uses its electricity output, along with grid electricity, to buffer the additional thermal energy demand. This operating strategy aims to maximize the utilization of the mCHP energy output.”

2.2 The results are impressive and the work support the conclusions well. But the advance of the system lies in the advanced OP4S engine rather than the whole mCHP system. The concept of mCHP systems based on waste heat recovery systems and energy storage systems has been widely reported. Please make a more clear novelty statement of the work.

Author response: The novelty statement of the mCHP work has been summarized and highlighted in the conclusion as below:

In lines 406–419: “A compact and portable mCHP prototype at TRL 6 has been developed with several novel features, including adoption of the highly efficient OP4S engine, a flexible fuel capability, and a well-designed waste heat recovery system capable of recovering two waste heat sources (i.e., exhaust and hot coolant) to provide hot water in meeting thermal load demand. The mCHP prototype enables up to 35.2% of fuel-to-electricity efficiency and nearly 93% of the overall mCHP efficiencies, exceeding the conventional mCHP’s fuel-to-electricity efficiency limit of 30%. Achieving high electrical efficiency is always critical and challenging in the development of novel mCHP technologies. Moreover, the OP4S engine has 60% fewer parts per engine unit, which, therefore, lowers material and manufacturing costs while enabling longer service time. The combination of high efficiency and cost-effectiveness enables significant potential deployment and market penetration for the mCHP in the US residential sector. In addition, the mCHP prototype can be powered by traditional fuels such as natural gas or propane but also has the potential to run on carbon-free fuels such as hydrogen. These features indicate the technology has a substantial potential of supporting the transition from current conventional fossil fuels to carbon-free fuels in the future. In addition, the mCHP is a compact and portable device, allowing high versatility for installation locations.”

We also provide a brief discussion in Supplemental note 20.

2.3 It is recommended to use a Sankey diagram to represent Figure 5, making the flow and distribution of energy more clear and comprehensible.

Author response: The Sankey diagrams for energy and exergy analysis of the system have been conducted and reported in the revision. Figure 5 has been updated to a Sankey diagram. Figure S9 shows the Sankey diagrams of exergy flow and destruction in each component. Supplemental Note 15 lists Sankey diagrams for energy and exergy analysis of all 10 cases. The detailed data for the system and component losses are also provided in Supplemental Note 15.

Reviewer #3:

3.1 The motivation for the proposed mCHP technology deployment in residential/”light commercial” buildings should be elaborated upon in the context of/relative to other competing low-carbon technologies, including renewable electricity and renewable heat supply. In addition to thermodynamic efficiency, the fuel used in the proposed mCHP technology, and its expected penetration in buildings, will determine potential emissions abatement globally in a given timeline. “Light commercial” buildings needs to be defined.

Author response: In lines 152–162, we address the following motivation for the proposed mCHP technology deployment in residential/light commercial buildings: “Overall, deploying the mCHP technology in residential and light commercial buildings can provide uninterrupted heat and power at high efficiency and low cost without the constricts of severe weather. This is important for cold-climate regions and remote communities, which frequently experience severe weather or weather-related disasters. The mCHP can also be coordinated with intermittent renewable energy sources (e.g., wind and solar) to provide power and heat when the renewable energy sources are not available or during extended grid outages. It can also serve as the backbone for renewable energy–based microgrids by providing a reliable baseload

source of electricity and thermal energy to support renewable energy resources and energy storage. By utilizing clean fuels such as hydrogen, the mCHP can achieve near-zero carbon emissions. In addition, the mCHP can be installed directly in buildings. Therefore, the mCHP is a compelling option for those seeking an uninterrupted, efficient, and cost-effective solution for both electricity and heat generation in residential and light commercial buildings.”

Regarding the reviewer’s comment on *“In addition to thermodynamic efficiency, the fuel used in the proposed mCHP technology, and its expected penetration in buildings, will determine potential emissions abatement globally in a given timeline,”* the authors agree with the point. This work aims at designing and developing an efficient and cost-effective mCHP technology, enabling a potential market penetration of mCHPs in building applications and a potential carbon footprint reduction. The market penetration of mCHPs in building applications is substantially affected by market dynamics and government subsidies and policies. Assessment of this uncertainty is currently challenging. Therefore, this is not discussed in this article because it is beyond the scope of the work.

Regarding the definition of *“light commercial”* buildings, a light commercial building refers to a type of structure that is designed for smaller-scale-operation business and commercial purposes compared with heavy commercial or industrial buildings. These buildings are typically used for retail stores, small offices, restaurants, and so on. It has been defined in lines 76–77.

3.2. Among the potential fuels listed (i.e., biogas, natural gas, propane or hydrogen), are natural gas and propane effective/viable fuel options considering global decarbonization goals – what is the anticipated timeline for the deployment of the proposed mCHP technology. The emission levels achievable with different fuels would need to be quantified, including upstream emissions in the case of fossil gas. As part of “performance” metrics, emissions other than GHG also need to be quantified. The fuel employed to generate the experimental/simulation results presented, and compare the performance of the proposed system (energy, emissions) with competing technologies/system(s) of reference, needs to be explicitly documented throughout the manuscript including as part of figures/tables of results; Line 106 indicates natural gas; can its composition be documented. How is thermodynamic performance expected to be influenced by fuel, and thus comparison of performance with competing system(s)/system(s) of reference.

Author response: We have separated our answer to this inquiry into five sections, detailed below.

Regarding the comment of *“Are natural gas and propane effective/viable fuel options considering global decarbonization goals? – what is the anticipated timeline for the deployment of the proposed mCHP technology?”*, we addressed this in lines 113–119: *“When considering global decarbonization goals, natural gas and propane are generally regarded as transitional or intermediate fuels rather than long-term sustainable options. However, global decarbonization will be a long-term effort. It is important for developing novel mCHP technologies to support the transition from current conventional fossil fuels to zero carbon emissions in the future. Thus, the proposed mCHP is optimally designed for natural gas but can also run renewable biogas and hydrogen. The anticipated timeline for the deployment of the proposed mCHP technology is targeted within 3–5 years.”*

Regarding the comment of *“The emission levels achievable with different fuels would need to be quantified, including upstream emissions in the case of fossil gas. As part of “performance” metrics, emissions other than GHG also need to be quantified,”* the authors have to politely mention that it will be a very heavy work to test and quantify all the detailed emission levels achievable with different fuels, including hydrogen, biogas, and natural gas. The studies for each fuel mentioned could be a couple PhD dissertations. The suggested work and scope are out of the scope presented in the paper. We only measured the emissions, including CO, HC, and NO_x, from the mCHP prototype using natural gas. The measurements show that the prototype meets US EPA new source performance standards (NSPSs) for emissions for a spark-ignition stationary engine used in the power generation of less than 19 kW. For the engine displacement of the

mCHP, the NSPSs require CO emissions of no more than 610 g/kWh and HC+NO_x emissions of no more than 8 g/kWh. The results are shown in the figures below. We addressed emissions of the mCHP in Lines 222–228, as well as Supplemental Note 14 in detail.

Lines 226–232, we addressed: “Additionally, emissions such as CO, HC, and NO_x were also measured for all the lean and stoichiometric modes. Detailed results are shown in Supplemental Note 14, which confirms that the prototype using natural gas meets US Environmental Protection Agency new source performance standards (NSPSs) for emissions for SI stationary engines used in the power generation of less than 19 kW. For engine displacement of the mCHP, the NSPSs require CO emissions less than 610 g/kWh and HC+NO_x emissions less than 8 g/kWh [50]. Compared with the stoichiometric modes, all the lean modes yielded substantially lower CO, HC, and NO_x emissions.”

Figure S8. (a) HC+NO_x emissions of all the lean and stoichiometric modes as a function of engine power, and (b) CO emissions of all the lean and stoichiometric modes as a function of engine power. The error bars shown are 10%.

Regarding the comments of “The fuel employed to generate the experimental/simulation results presented, and compare the performance of the proposed system (energy, emissions) with competing technologies/system(s) of reference, needs to be explicitly documented throughout the manuscript including

as part of figures/tables of results” and “How is thermodynamic performance expected to be influenced by fuel, and thus comparison of performance with competing system(s)/system(s) of reference,” Figure 6(a) compares the AC electrical efficiency of the mCHP prototype with other mCHP systems. It is evident that when the mCHP operates beyond 4 kW, its efficiency surpasses that of other competing systems, whereas it maintains a marked advantage over them at 5 kW and 6 kW operations owing to the innovative mCHP powered by OP4S engine. Figure 6(b) further highlights that the overall mCHP efficiency is comparable to that of the competing systems. Unfortunately, it is hard to have the emissions data from these competing systems. Thus, we did not compare the emissions between the developed mCHP and the competing systems but rather validated that the developed mCHP meets current emissions standards.

Regarding the comment of “Line 106 (in original version) indicates natural gas; can its composition be documented,” the composition of the fuel used is added in the revised supplemental document file. For the details, please see Supplemental Note 5.

Table S6. Fuel composition of natural gas used in the testing cases shown in Figure 4.

Nitrogen	0.1953%
Methane	91.7073%
Ethane	7.4238%
Propane	0.2544%
Butane	0.0178%
Isobutane	0.0208%
Pentane	0.0012%
Isopentane	0.0036%
Hexanes	0.0029%
Others	0.3729%
Lower heating value (kJ/mol)	848
Higher heating value (kJ/mol)	939
Carbon	75.7%
Molecule weight (g/mol)	17.0

3.3 Experimental measurement uncertainty, tested repeatability, reproducibility require to be further documented. Experimental uncertainty needs to be visualized in figures/tables of results to make the presentation of the results more self-explanatory.

Author response: In lines 175–181, we address: “In the tests, the measurement uncertainty and testing repeatability were analysed. The results show that the uncertainties of electrical, thermal, and total CHP efficiency are 3.28%, 2.26%, and 2.51%, respectively (see Supplemental Note 3). The sensitivity comparison of the repeated tests for a 5.93 AC kW lean combustion mode was also conducted (see Supplemental Note 4), and the observation shows less than 1.8% error for all the performing efficiencies, power, OP4S exhaust, and coolant. Thus, the mCHP can achieve stable and repeatable operation.”

We add the details in Supplemental Note 3 and Table S4, shown as below:

“The propagation of the uncertainty method [R1] was used to calculate the uncertainty associated with the measurements in the work. The method estimates the uncertainty in a parameter from the uncertainties in the measurements used to calculate the parameter. For example, the uncertainty of energy in the burnt fuel depends on the uncertainties of fuel flow measurement and lower heating value (LHV) calculation in the

testing cases. In the same manner, the uncertainty of total efficiency of the system is a function of measurement uncertainties in temperature, pressure, and mass flow rates. The key instruments involved in the measurements for the calculation of mCHP efficiencies are listed in Table S4. In the current system, the uncertainties of electrical, thermal, and total efficiency are derived as 3.28%, 2.26%, and 2.51%, respectively. All the experimental uncertainty has been shown in Figure 4 in the revised manuscript.”

Table S4. Measurement range and sensitivity of instruments used in the tests.

Instrument description	Measurement range	Sensitivity
K-type thermocouple	−200 °C to 1250 °C	2.2 °C
Pressure transducer	Different ranges for different applications, such as in-cylinder, oil, water, and manifold	2.1%
Coolant flow meter	0.0–6.6 GPM	0.8% of the measured value +0.5% of the final value of the range
Gas flow meter	1–120 SFM	1%
Voltage transducer	0–500 V	1%
Current transducer	0–300 A	1%
Oxygen sensor	20–900 mV	1–45 ppm

Regarding tested repeatability and reproducibility, we repeated the test for the ~6.0 AC kW of the lean combustion modes (i.e., the cases 9 and 10). The repeatability results are detailed in Tables S5 and Figure S1 in Supplemental Note 4. Tables S5 and Figure S1 are attached below. The maximum sensitivity is less than 1.8% for all the performing efficiencies, power, OP4S exhaust, and coolant. This indicates that the mCHP is capable of excellent repeatability and reproducibility.

Table S5. Repeatability and reproducibility results for the mCHP.

Parameter	Case	Waste heat recovered and stored in the tank		Repeatability resolution
		9	10	(Based on case 9)
AC electrical power (kW)		5.93	5.98	1.8%
Engine brake energy efficiency (%)		39.1	38.7	−1.0%
OP4S-out coolant temperature (°C)		79.0	79.4	0.5%
OP4S exhaust temperature (°C)		692	699	1.0%
AC electrical power efficiency (%)		35.2	34.8	−1.1%
DC electrical power efficiency (%)		33.8	33.4	−1.2%
Overall mCHP efficiency (%)		93.2	93.6	0.4%

Figure S1. Repeatability resolution analysis for cases 9 and 10.

3.4. The system flexibility with regard to power to heat ratio to meet the needs of different (building) users would benefit from more explicit documentation of the power-to-heat ratios and their effect on thermodynamic performance. Full load versus part load performance also requires more explicit documentation as part of the results.

Author response: In the revision, we have addressed the question of “*The system flexibility with regard to power to heat ratio to meet the needs of different (building) users*” in answering Q2.1.

Regarding “*Full load versus part load performance,*” the manuscript has clearly demonstrated the mCHP performance over the range of 3.7–7.4 DC kW (or 3.9–7.7 AC kW) under stoichiometric modes and over the range of 3.5–5.8 DC kW (or 3.6–6.0 AC kW) under lean modes. The reduced maximum power delivery in the lean mode is due to less fuel available for combustion. In a lean-burn mode, the air-fuel mixture contains a higher proportion of air (i.e., rich oxygen) compared with the stoichiometric mode. Instead, the stoichiometric mode ensures that the engine receives the ideal amount of fuel for combustion, resulting in maximum power output. The system can be flexible to set full load versus part load based on users’ energy needs and is capable of handling dynamic load profiles.

3.5. Analyzing and reporting second-law based thermodynamic performance would provide additional insight into the performance of the proposed mCHP technology in the different test configurations/cases considered, and how it compares with system(s) of reference (e.g., Cummins system, systems in Table S1).

Author response: We conducted the exergy analysis for the proposed mCHP and have explained it in Q1.3. Regarding the systems of reference (e.g., Cummins system and systems in Table S1), their detailed

information for exergy deterioration analysis is not released in the public domain. It is challenging to evaluate these mCHPs without that information. However, we will keep this in mind and have the data published in the future once it is made available from collaborators.

3.6. CFD simulations results are presented focusing on the water storage tank. CFD model validation (i.e., comparison between CFD and corresponding experimental results) should be presented. The objectives of CFD simulation should be further clarified in the context of the experimentally focused work, i.e. contribution of the simulation results to the overall manuscript analysis.

Author response: In the revision, the validation of the CFD model for the water tank is added in the supplemental document (see Supplemental Note 10). The objective of the CFD simulation was to help the understanding of the performance of water heating and support the optimal coil design for water heating in the future. However, we can remove it if the reviewer doesn't consider it useful for potential readers.

3.7. Have experimental results presented in the manuscript been compared with corresponding thermodynamic model predictions, which could be useful to extend the results presented to other test configurations.

Author response: The current work focuses primarily on the development and testing of novel micro-CHP. Although we developed an OP4S thermodynamic model using the GT suite in limited time, the model is insufficient or limited to reflect the complicated physical behaviours, such as combustion and fluid flow movements, in the OP4S engine. Thus, the mode and results were used only to understand the performing trend of the OP4S engine. We did not compare the micro-CHP experimental results with the thermodynamic model predictions.

3.8. Maintenance, reliability, lifetime, safety, recyclability and cost are other aspects for application of the proposed mCHP technology to buildings, which could be (further) addressed in the manuscript.

Author response: In the revision, we address the maintenance, reliability, lifetime, safety, recyclability, and cost for application of the proposed mCHP.

In lines 412–413, we added: *“Moreover, the OP4S engine has 60% fewer parts per engine unit, which, therefore, lowers material and manufacturing costs while enabling longer service time. .”*

In lines 133–134, we added: *“The OP4S engine is designed with a 20-year life span because of its excellent reliability and durability owing to substantially fewer parts than other engine types and low vibration [46].”*

In lines 252–256, we added: *“The mCHP has been tested for more than 600 hours without damage or deterioration, indicating its high reliability and safety. The unit has the capability to replace a residential furnace, a water heater, and grid power supply. The unit is recommended for annual replacement of oil and oil filter, spark plug, and air filter, as well as lubrication of the water pump. The detailed cost is shown in Supplemental Note 18: Table S16.”*

In addition, in the revision, we conducted a life cycle analysis for the proposed mCHP in building applications. The analysis accounts for annual operation and maintenance cost savings, initial investment penalty, and disposal cost penalty at the end of its life cycle. For the details, please see (1) lines 495–521, Methodology of Economic Analysis and Decarbonization; (2) The details of Methodology of Economic Analysis and Decarbonization, Supplemental Note 18; and (3) lines 343–403, Discussion of the Cost Savings and Decarbonization of mCHP.

3.9. The main general comment is concerned with the significance of the manuscript contents for possible publication. If the proposed technology is already commercialized, the potential novelty and advancement contributed by the manuscript submission require to be clarified and further highlighted in the manuscript

submission. The analysis methods employed in the manuscript (experimental/modelling-simulation) are not considered novel or state of the art (including based on above comments). The results (Fig. 6) indicate that the proposed system can reach high-bound mCHP efficiencies; what would be globally the emissions saved by the proposed technology in what timeline considering its expected penetration in buildings; would the proposed technology be considered a key technology to achieve net-zero emissions energy systems by mid century.

Author response: The mCHP technology presented in this paper is still a verified prototype system with TRL 6/7 and has been demonstrated in a laboratory operational environment. Potential novelty and advancement contributed by the developed mCHP has been addressed in the conclusion and Q2.2.

Regarding decarbonization, the carbon intensity of grid electricity is the key factor in determining the value of the proposed system as a decarbonization tool. The regions with highly polluting electrical grids will benefit the most if the mCHP is employed today. This has been discussed in lines 377–379. Moreover, the application of green hydrogen in the mCHP will achieve complete decarbonization. Therefore, the developed mCHP can support the transition from current conventional fossil fuels to zero carbon emissions in the future.

The detailed techno-economic analysis reported in the main manuscript (Fig. 7 and 8) provides the economic and environmental benefits of the technology in regions with different grid carbon intensities. Although the chosen regions' carbon intensity is in the moderate range relative to the global electric grid profiles, the savings are anticipated to extrapolate and be applicable in regions beyond the US, particularly in cold climates of the world. Considering the paper to be focused on development of the mCHP technology, we did not evaluate the technology's global GHG reduction with respect to its expected penetration in buildings, which is out of the scope in the work. Many works have evaluated global GHG reduction using CHP applications in buildings (see the introduction).

3.10 specific comments

3.10.1 Manuscript submission title: The title does not refer to buildings as the proposed mCHP technology application. Could the manuscript submission title be revised accordingly; otherwise could this omission in the existing title be in the existing title justified.

Author response: In the revision, we have revise the manuscript title as “Development of an Advanced Micro-CHP Powered by an Opposed-Piston Engine in Building Applications”. We thank the reviewer's suggestion.

3.10.2 Abstract: The fuel and analysis methods require to be documented in the abstract, as well as measured/modeled GHG emissions savings relative to competing technology(ies)/system(s) of reference. The last sentence in the abstract needs to be supported by evidence presented in the manuscript. The abstract does not refer to economics.

Author response: In lines 5–18, we rewrote the abstract by adding fuel, decarbonization, and cost savings potential analysis: “A prototype micro-combined heat and power (mCHP) system powered by an innovative opposed-piston engine was developed to simultaneously generate electricity and provide heat to residential homes or light commercial buildings. The mCHP prototype includes an inwardly opposed-piston four-stroke engine, generator, rectifier, inverter, battery energy storage system, 52-gal water tank, and waste heat recovery accessories for hot water supply and space heating. The developed prototype can operate under lean and stoichiometric combustion modes. It attains the maximum AC electrical efficiency of 35.2%, and the engine brake thermal efficiency under the lean combustion mode approaches 40%. The exceptional electrical efficiency breaks the typical upper boundary of 30% for small internal combustion engine-based mCHP (i.e., <10 kW). Moreover, the mCHP enables maximum combined electrical and thermal efficiencies greater than 93%. The mCHP is optimally designed for natural gas but can also run renewable biogas and hydrogen, supporting the transition from current conventional fossil fuels to zero carbon emissions in the future. The decarbonization and cost-saving potential of mCHP were studied for 10 representative

residential houses. The results indicate that, except for specific locations, the mCHP excels in achieving decarbonization and cost savings primarily in US northern and middle climate zones. ”

The last sentence in the original abstract is removed in the revision.

3.10.3 Lines 33-35: What is the purpose of the statement; for example what geographical area(s) is(are) targeted for application in the manuscript submission. Should specific climate or socio-economic aspects be clarified.

Author response: In lines 33–35 (original version), the paper wants to address that substantial works on economic benefit, marketing, and residential applications of mCHP were performed in the world, including Asia, Europe, and the Middle East, This indicates a significant potential market for the application of mCHP in the building sector. In the revision, we updated the statement in lines 36-39 as below.

“Substantial work has been performed to understand the economic benefit, marketing, and residential applications of mCHP in Asia [6–8], Europe [9, 10], the Middle East [11], and North America [12–14]. There is a significant potential market demand for mCHP applications in the building sector. ”

Regarding “what geographical area(s) is(are) targeted for application in the manuscript submission. Should specific climate or socio-economic aspects be clarified,” we conducted a deep analysis on the effect of the mCHP system in all regions of the US. The presented economic analysis demonstrates the viability of mCHP in US northern and middle climate zones. Specific locations with lower thermal energy needs and clean grids are not suitable from an environmental benefit standpoint, but economic benefits can be realized in most locations examined. The details are shown in the section “Discussion of the Cost Savings and Decarbonization of mCHP.” On the other hand, the mCHP can also play a critical role in coordination with the renewable energy sources to provide power and heat in a resilient manner. It can serve as the backbone for renewable energy–based microgrids in combination with solar, wind, and energy storage to support baseload requirements (both electrical and thermal) of the buildings. This has been addressed in Q3.1. In addition, the novel mCHP is also expected to assist remote and underserved communities, which face significant challenges owing to vulnerable energy infrastructure and lack of access to reliable electricity. It is addressed in lines 80–85 on page 2.

3.10.4 Lines 54-70 (original version): The timeline of development of the proposed mCHP technology and current/future TRL need to be documented.

Author response: The proposed mCHP technology is currently at TRL 6/7 and is expected to reach the commercial market within 3–5 years. We addressed it in lines 118–119 of the novel mCHP development and in the conclusion at line 435.

3.10.5 Line 72 (original version): “small” in “a small OP4S engine” is to be quantified.

Author response: In the revision, we quantified the small OP4S engine in lines 75–77. The details are listed below:

“This work presents the development of a novel mCHP prototype powered by a small OP4S engine (i.e., <10 kW) to simultaneously provide heat and electricity to single-family houses or light commercial buildings.”

3.10.6 Lines 76-79 (original version): The potential suitability of the proposed mCHP technology in buildings needs to be further elaborated based on additional criteria, in the introduction and results/discussion sections of the manuscript submission.” Light-commercial buildings” needs to be defined (if possible, quantitatively).

Author response: In the abstract and introduction, we address the potential suitability of the proposed mCHP technology in building sectors including single-family houses or light commercial buildings. A light

commercial building refers to a type of structure that is designed for smaller-scale operation business and commercial purposes compared with heavy commercial or industrial buildings. These buildings are typically used for retail stores, small offices, restaurants, and so on. In the revision, a definition for light commercial building is provided in lines 76–77.

3.10.7 Line 77 (original version): Is “low cost” an outcome of the analysis documented in the submission? If not, it should be quantified/defined with appropriate citations.

Author response: In lines 62–63, we added, “Moreover, OPEs are more cost-effective than conventional ICEs, mainly because they have 60% fewer parts per engine unit [37] and, therefore, have lower material and manufacturing costs.” The reference [37] on the cost effectiveness of OPEs is added in the References list. We believe that the statement and reference support the definition of “low cost” well.

3.10.8 Lines 77-79 (original version): The following text requires further elaboration (i.e., drop in, connections, vulnerable energy infrastructure and high energy cost) in “drop-in replacement feature using the building’s existing connections, particularly in remote and underserved communities which faces significant challenges due to vulnerable energy infrastructure and high energy cost”. Please consider incorporating examples to support this claim.

Author response: In the revision of lines 80–85, we rewrote the sentences as, “*The novel mCHP technology is expected to promote mCHP acceptance in the residential and light commercial markets because of its low cost and drop-in replacement feature using the building’s existing connections. This is particularly important in remote and underserved communities that face significant challenges because of vulnerable energy infrastructure and a lack of access to reliable electricity. Such remote and underserved communities include rural villages in developing countries, mountainous regions, disaster-affected areas, and nomadic or transient communities.*”

3.10.9 Line 115 (original version): Please quantify.

Author response: In the revision of lines 130–133, we quantify the efficiency benefits as “Unlike conventional ICEs, which route the substantial heat of combustion to the cylinder head, the heat of combustion in the OP4S goes only to the opposing piston, reducing heat loss and increasing efficiency by 30%–50% more than that of comparable conventional petrol and diesel engines [37].”

3.10.10 Figure 6: Please specify fuel for all devices considered, and whether competing systems data is directly from manufacturers and the operating condition considered (e.g., full load/design conditions). Are all efficiency definitions used by the different cited sources to quantify device performance consistent.

Author response: In Supplemental Note 16, Table S14 is added to address the fuel type of the data shown in Figure 6. All the data in Figure 6 are cited directly from OEM datasheets or cited references. The data includes full and partial load conditions provided from OEM datasheets. All the AC electrical efficiencies are defined based on fuel lower heating value, and all the overall mCHP efficiencies are defined based on fuel higher heating value.

Table S14. Fuel types of the data shown in Figure 6.

mCHP model	Reference	Fuel type
OPS (ICE)	—	Natural gas
COGEN Microsystems (ORC)	[3]	Natural gas
Energetix: Genlec (ORC)	[3]	Natural gas
JX Crystal Prototype (TPV)	[3]	Natural gas
Senertec: Dachs (ICE)	[3,13,21,23]	Natural gas
Solo: Solo 161 (Stirling)	[3,13,22,23]	Natural gas
Honda: Ecowill (ICE)	[3,19]	Natural gas
Whisper Tech (Stirling)	[13,20]	Natural gas
Academic prototype (ORC)	[16]	Biogas
Academic prototype (TPV)	[17]	Natural gas
Stirling Denmark: SM5A (Stirling)	[23]	Natural gas
EC POWER: XRGI® 6 (ICE)	[24]	Natural gas, propane, butane
EC POWER: XRGI®9 (ICE)	[25]	Natural gas, propane, butane
AEG: Ecopower (ICE)	[26]	Natural gas, propane
TOTEM: TOTEM 10 (ICE)	[27]	Methane
YANMAR: CP5WN (ICE)	[28]	Natural gas

In the manuscript revision: lines 263-267, we updated as, “The results are shown in Figure 6, and the fuel type of each model displayed in the figure is shown in Supplemental Note 16. The AC electrical efficiencies are evaluated based on fuel lower heating value (LHV), and the overall mCHP efficiencies are evaluated based on fuel higher heating value (HHV) because a portion of the latent energy in water vapor is usually recovered.”

3.10.11 Line 253 (original version): How are the emissions savings and cost reductions estimated (methods including assumptions and calculations). In Figure 7, could a legend be included to define colour coding. Including % CO₂ emissions and % cost savings could be valuable. The figure text font graphical quality requires improvements.

Author response: The methods, including assumptions and calculations, for evaluating carbon emissions savings and cost reductions are detailed in Supplemental Note 18. Figure 7 has been updated by adding a legend, CO₂ reduction, and cost savings percentage. We appreciate the reviewer’s suggestions. Figure 8 is added to account for life cycle CO₂ reduction and cost savings.

Figure 7. Effect of mCHP applications on annual (a) decarbonization and (b) cost savings of the selected 10 households, which are in Anchorage (Alaska), Boston (Massachusetts), Detroit (Michigan), Minneapolis (Minnesota), Boulder (Colorado), Kansas City (Missouri), Lexington (Kentucky), Atlanta (Georgia), Fort Worth (Texas), and Los Angeles (California). The colours marked in the figure show annual CO₂ emissions and operation cost levels of the selected homes.

Figure 8. Life cycle decarbonization and cost savings of mCHP applications in the selected 10 households. The life span of mCHP is 20 years.

3.10.12 Conclusions, Lines 273-274 (original version): The following text “can use a variety of fuels”, “can provide flexible and thermal electrical loads”, “the OP4S is simpler and cheaper to manufacture, assemble and operate” are not found to be supported by analysis presented in the manuscript. If such information was sourced and demonstrated elsewhere, it would be more appropriately placed in earlier parts of the manuscript (e.g., introduction) with cited references.

Author response: We appreciate the reviewer’s suggestions. We substantially revised the manuscript to address the question through adding appropriate content in appropriate parts of the manuscript.

Regarding “can use a variety of fuels,” we have addressed it in Q.3.2 and discussed renewable fuel effects on the OP4S and mCHP system in lines 305–343.

Regarding “can provide flexible and thermal electrical loads,” we have addressed it in Q.2.1 in detail.

Regarding “the OP4S is simpler and cheaper to manufacture, assemble and operate,” we have addressed it in lines 412–413 as, “Moreover, the OP4S engine has 60% fewer parts per engine unit, which, therefore, lowers material and manufacturing costs while enabling longer service time.”

3.10.13 Conclusions, Line 277 (original version): “if appropriate technologies can be applied to overcome hardware limitations” is vague. What are the technologies and hardware limitations referred to.

Author response: We have rewritten the conclusion. This content has been comprehensively discussed in the section on Discussion of mCHP Efficiency. It mentions that the 40% AC electrical efficiency target could be achieved through the following further approaches: (1) a higher compression ratio (e.g., 16 or above); (2) an even leaner combustion of an equivalent ratio of 0.6; (3) an implementation of Miller cycle application with a higher expansion ratio; (4) more efficient generator with 95% efficiency; and (5) a compact ORC device used to recover the exergy loss in the OP4S exhaust flow, which is addressed in lines 285–292. To avoid any vague discussion, the sentence is removed in the revision’s conclusion.

3.10.14 Conclusions, Line 278: the methods employed for calculating emissions savings need to be elaborated in the manuscript. In general, decarbonization potential should consider not only specific emissions savings but also the expected technology penetration potential in buildings. No economic analysis is found in the manuscript to support “..cost savings potential were studied”.

Author response: The main effort in the work is to develop an efficient and cost-effective mCHP. The authors agree that it is important to evaluate the emissions savings of the expected technology penetration potential in buildings, but it is out of the scope of the work.

In the revision, the life cycle economic analysis is conducted. The details can be found in lines 343–403, as well as Supplemental Notes 18 and 19. Supplemental Note 18 shows the method of life cycle economic and CO₂ emissions analysis.

3.11 Supplementary information document:

3.11.1 In order to be more self-explanatory, figures of results (such as Fig. S1) and tables of results should specify the fuel composition used in the proposed mCHP technology.

Author response: The fuel composition has been shown in Q3.2.

3.11.2 In Table S1(original version), what are the intended applications of the competing mCHP systems listed. Are the electricity/heat outputs and electrical/overall energy efficiencies obtained directly from mCHP vendors and in what conditions (e.g., part and full load/design conditions). Is the temperature level of the heat output available. Comments on the main reasons for differences in thermodynamic performance between reported systems would be informative. Are vendor data obtained using specific procedures/standards.

Author response: Table S1 summarizes the competing mCHP systems available in the worldwide market because it is helpful to the audience in understanding the R&D situation on mCHP technologies. All the electrical/overall energy efficiencies are cited from the mCHP vendors and the publications. The references show the very detailed load conditions. To avoid a lengthy paper, we did not list all such information in the manuscript and supplemental document.

In Supplemental Note 1: lines 4-8, we added the following comment for differences in thermodynamic performance between reported systems: *“Clearly, their performance is substantially different. This considerable variation in performance is a direct result of employing different mCHP technologies, design and fabrication approaches, and system integration and optimization considerations, all while considering cost-effectiveness. Additionally, specific procedures and standards in different countries may also contribute to these differences.”*

3.11.3 Table S2/S3(original version): Additional information on controls for water outlet temperature.

Author response: In Supplemental Note 2: lines31-27, we added *“In compliance with ASHRAE standards, water heaters are required to supply hot water at a temperature of 60 °C (140 °F). Therefore, the case 1–2 and 7–10 are designed to raise water temperature from 26.7 °C to 60 °C without external heat load demand. In the case 3, the water within the tank is heated from 61.1 °C to 71.1 °C without external heat demand. Additionally, the case 4–6 are designed because typical US domestic space heating temperature, generated from heat pumps and residential furnaces, is in the range of 21.1 °C to 51.7 °C (70 °F to 125 °F). These cases have the water tank beginning at 40 °C and incorporating external thermal demand for space heating requirements.”*

3.11.4 In Tables S6 and S7(original version), including the available waste heat and percent waste heat recovery (relative to available waste heat) may be of interest. The electrical and thermal efficiencies could also be included.

Author response: In Tables S9 and S10 in the revised Supplemental Note 7, the available waste heat from exhaust and coolant and the percentage of waste heat recovery were added. The electrical and thermal efficiencies have been listed in Tables S7 and S8 in the revised Supplemental Note 6.

3.11.5 Concerning the comparison between the proposed mCHP system and system of reference (e.g., Cummins system), for each test configuration (case), in order to provide additional insight, electrical and thermal efficiencies, overall energy efficiencies, as well as second law efficiencies, would require to be compared.

Author response: The question has been addressed in Q3.2 and Q3.5. For the details, please see our answers to Q3.2 and Q3.5.

REVIEWERS' COMMENTS

Reviewer #2 (Remarks to the Author):

The revised manuscript has addressed the issues. One more suggestion, This article evaluates the performance of the CHP system using total energy efficiency, which is based on the first law of thermodynamics and does not consider the difference in quality between electrical and thermal energy. It would be better to add the evaluation parameters such as heat-electricity ratio equivalent electrical efficiency for more comprehensive evaluation.

Reviewer #3 (Remarks to the Author):

The authors have provided sufficient responses to previous review comments on the initial manuscript submission, with the majority of reviewer suggestions implemented in the revised manuscript submission, which has been enhanced.

General response

We thank the reviewers' time and effort in reviewing the revised manuscript. Below, you will find the original reviewer comment, followed by our response. Hopefully our response answer the question.

Reviewer #2:

2.1 The revised manuscript has addressed the issues. One more suggestion, This article evaluates the performance of the CHP system using total energy efficiency, which is based on the first law of thermodynamics and does not consider the difference in quality between electrical and thermal energy. It would be better to add the evaluation parameters such as heat-electricity ratio equivalent electrical efficiency for more comprehensive evaluation.

Author response: We agree it is very important to have the quality of energy accounted for in evaluating the performance of CHP system. In the final version, we added both AC-based and DC-based exergy electrical efficiencies in the Table S12 of Supplemental Note 15: Exergy analysis. Table S2 is attached as below. Additionally, the table also shows the AC-based or DC-based total mCHP exergy efficiencies, which include water tank exergy recovered.

In the manuscript Line 273-276, we address: “More results on exergy analysis are provided in Supplemental Note 15, where Table S12 shows exergy electrical efficiencies and total mCHP exergy efficiencies. Unlike exergy electrical efficiencies, the total mCHP exergy efficiencies account for very limited water tank exergy recovered besides power generation.”

Table S12. Summary of exergy flow and destruction of the 10 cases.

Case	Case 1	Case 2	Case 3	Case 4	Case 5	Case 6	Case 7	Case 8	Case 9	Case 10
Combustion mode	Stoich	Stoich	Stoich	Stoich	Stoich	Stoich	Lean	Lean	Lean	Lean
DC power (kW)	3.74	4.42	7.40	3.20	4.50	5.94	3.50	4.43	5.70	5.74
AC power (kW)	3.90	4.60	7.71	3.33	4.69	6.18	3.65	4.61	5.94	5.98
Water tank exergy recovered (kW)	0.81	0.87	2.37	0.71	1.176	0.74	0.87	0.72	0.77	0.67
Exergy electrical efficiency, DC (%)	15.4	16.5	24.2	15.4	16.8	23.5	20.9	27.3	32.4	32.0
Exergy electrical efficiency, AC (%)	16.1	17.2	25.2	16.0	17.5	24.5	21.8	28.5	33.8	33.4
Total exergy efficiency, DC (%)	18.8	19.8	32.0	18.8	21.2	26.5	26.0	31.8	36.8	35.8
Total exergy efficiency, AC (%)	19.4	20.4	33.0	19.5	21.9	27.5	26.9	32.9	38.1	37.1
Exhaust loss, destroyed (kW)	0.58	0.66	1.12	0.81	0.66	0.88	0.38	0.37	0.40	0.41
Engine heat loss, destroyed (kW)	0.18	0.20	0.24	0.16	0.20	0.22	0.13	0.13	0.17	0.15
Engine, destroyed (kW)	14.43	15.80	15.12	11.00	15.23	9.65	8.44	7.25	7.01	7.30
Generator, destroyed (kW)	0.43	0.51	0.856	0.37	0.52	0.69	0.41	0.51	0.66	0.66
Rectifier, destroyed (kW)	0.16	0.18	0.31	0.13	0.19	0.25	0.15	0.18	0.24	0.24
Water tank heat loss, destroyed (kW)	0.00	0.00	0.00	0.00	0.00	0.00	0.00	0.00	0.00	0.00
Water tank, destroyed (kW)	3.92	4.13	3.135	4.38	4.309	6.86	2.90	2.60	2.63	2.73
Total exergy loss (kW)	19.71	21.49	20.78	16.85	21.12	18.55	12.40	11.06	11.11	11.50
Generator exergy input (kW)	4.33	5.12	8.56	3.70	5.21	6.87	4.05	5.13	6.60	6.64
Rectifier exergy input (kW)	3.90	4.60	7.71	3.33	4.69	6.18	3.65	4.61	5.94	5.98

Engine-out exhaust exergy (kW)	4.90	5.57	5.82	5.48	5.57	7.96	3.63	3.21	3.39	3.50
Engine-out coolant exergy (kW)	0.41	0.09	0.80	0.42	0.58	0.54	0.52	0.48	0.41	0.32
Fuel exergy input (kW)	24.25	26.78	30.55	20.76	26.80	25.23	16.77	16.20	17.58	17.91